# SLaM: Student-Label Mixing for Distillation with Unlabeled Examples

**Vasilis Kontonis**
UT Austin
vasilis@cs.utexas.edu

**Fotis Iliopoulos**
Google Research
fotisi@google.com

**Khoa Trinh**
Google Research
khoatrinh@google.com

**Cenk Baykal**
Google Research
baykalc@google.com

**Gaurav Menghani**
Google Research
gmenghani@google.com

**Erik Vee**
Google Research
erikvee@google.com

## Abstract

Knowledge distillation with unlabeled examples is a powerful training paradigm for generating compact and lightweight student models in applications where the amount of labeled data is limited but one has access to a large pool of unlabeled data. In this setting, a large teacher model generates "soft" pseudo-labels for the unlabeled dataset which are then used for training the student model. Despite its success in a wide variety of applications, a shortcoming of this approach is that the teacher's pseudo-labels are often noisy, leading to impaired student performance. In this paper, we present a principled method for knowledge distillation with unlabeled examples that we call Student-Label Mixing (SLaM) and we show that it consistently improves over prior approaches by evaluating it on several standard benchmarks. Finally, we show that SLaM comes with theoretical guarantees; along the way we give an algorithm improving the best-known sample complexity for learning halfspaces with margin under random classification noise, and provide the first convergence analysis for so-called "forward loss-adjustment" methods.

## 1 Introduction

While good quality human-labeled data are often hard to obtain, finding huge amounts of unlabeled data is relatively easy. Therefore, in modern machine learning applications, we often face the situation where we have a small "golden" dataset with human labels and a large unlabeled dataset. In *Distillation with Unlabeled Examples* [12, 26, 16] a large teacher model is first trained (or fine-tuned) on the human-labeled data and is then used to generate "soft" *pseudo-labels* for the unlabeled dataset. Then the (typically smaller) student model, i.e., the model that will be deployed for the purposes of the application, is trained on the combined dataset that contains both the labels generated by humans and the pseudo-labels generated by the teacher model. This general-purpose training paradigm has been applied in a wide variety of contexts [16, 46, 53, 54, 58, 57] including but not limited to distilling knowledge from large-scale foundational models like BERT [18] and GPT-3 [11]. We remark that in such settings one does not have access to the teacher model but only on its pseudo-labels (which were generated during some previous "bulk-inference" phase). This "bulk-inference" step is typically computationally expensive and happens once: one cannot modify the teacher network (or even use it for inference) during the training process of student.

Despite its widespread success in practice, the effectiveness of this powerful approach generally depends on the quality of the pseudo-labels generated by the teacher model. Indeed, training the student model on noisy pseudo-labels often leads to significant degradation of its generalization

37th Conference on Neural Information Processing Systems (NeurIPS 2023).

performance, and this is a well-known phenomenon that has been observed and studied in a plethora of papers in the literature, e.g., [6, 36, 44, 51, 53, 8, 27].

In this work, we propose Student-Label Mixing (SLaM), a principled method for knowledge distillation with unlabeled examples that accounts for the teacher's noise and consistently improves over prior approaches. At the heart of our method lies the observation that the noise introduced by the teacher is neither random nor adversarial, in the sense that it correlates well with metrics of "confidence" such as the margin score or the entropy of the teacher's predictions. We exploit this empirical fact to our benefit in order to introduce a model for the teacher's noise, which we use to appropriately modify the student's loss function. At a high level, for any given example during the student's training process, we evaluate the student's loss function on a convex combination of the student's current prediction and another (soft-)label that we estimate using our model for the teacher's noise (hence the name "student-label mixing").

Our contributions can be summarized as follows:

1. We propose SLaM: a principled method for improving knowledge distillation with unlabeled examples. The method is efficient, data-agnostic and simple to implement.

2. We provide extensive experimental evidence and comparisons which show that our method consistently outperforms previous approaches on standard benchmarks. Moreover, we show that SLaM can be combined with standard distillation techniques such as temperature scaling and confidence-based weighting schemes.

3. We give theoretical guarantees for SLaM under standard assumptions. As a byproduct of our analysis we obtain a simple "forward loss-adjustment" iteration that provably learns halfspaces with $\gamma$-margin under Random Classification Noise with $O(1/(\epsilon^2 \gamma^2))$ samples improving over prior works that had worse dependence on either the margin $\gamma$ or the generalization error $\epsilon$ (see Theorem 5.1 and Remark 5.2).

## 2   Related Work

**Knowledge Distillation.** Most of the literature on knowledge distillation has been focused on the *fully supervised/labeled setting*, i.e., when distillation is performed on the labeled training data of the teacher model rather than on new, unlabeled data — see e.g. the original paper of [26]. Naturally, in this setting the pseudo-labels generated by the teacher are almost always accurate and so many follow-up works [2, 14, 15, 41, 52] have developed advanced distillation techniques that aim to enforce greater consistency between the teacher's and the student's predictions, or even between the intermediate representations learned by the two models. Applying such methods in our setting where the training dataset contains mainly unlabeled examples is still possible but, in this case, it is known [51, 27] that fully trusting the teacher model can be actually harmful to the student model, making these methods less effective. (In fact, when the teacher is highly noisy these methods even underperform vanilla distillation with unlabeled examples.) In Section 4.2 we present results that show the improved effectiveness of SLaM relative to the state-of-the-art supervised knowledge distillation methods like the Variational Information Distillation for Knowledge Transfer (VID) framework [2]. Moreover, in Appendix D.5 we show that our method can be combined with (i.e., provide an additional improvement) the most simple, yet surprisingly effective, methods of improving knowledge distillation, namely the temperature-scaling idea introduced by [26].

For distillation with unlabeled examples, many approaches [17, 33, 29] propose filtering-out or reweighting the teacher's pseudo-labels based on measures of teacher's uncertainty, such as dropout variance, entropy, margin-score, or the cut-statistic. These methods are independent of the student model and can be synergistically combined with our technique. For instance, in Section D.4 we demonstrate that combining our method with teacher-uncertainty-based reweighting schemes leads to improved student performance relative to applying the reweighting scheme alone.

Much more closely related to our approach is the recently introduced approach of [27]. There, the authors design a model for the teacher's noise and utilize it in order to modify the student's loss function so that, in expectation, the loss simulates the loss with respect to noise-free pseudo-labels. One of the main advantages of our method compared to that of [27] is that our model for the teacher's noise is more structured and easier to learn, which — as our experiments in Section 4.2 show — leads to consistently better student performance.

**Learning From Noisy Labels.** Learning from noisy labels is an important and well-studied problem with a vast literature [7, 21, 23, 28, 31, 37, 40, 42, 45, 47] — see [50] for a recent survey. The fundamental difference between our setting and papers in this literature is that the noise introduced by the teacher is structured, and this is a crucial observation we utilize in our design. Specifically, our approach is inspired by the so-called *forward loss-adjustment methods*, e.g. [43], but it is specifically tailored to the structure of the distillation with unabeled examples setting. Indeed, forward methods typically attempt to estimate a noise transition matrix whose $(i, j)$ entry is the probability of the true label $i$ being flipped into a corrupted label $j$, which can be rather problematic when dealing with general, instance specific noise like in the case of distillation with unlabeled examples. On the other hand, we exploit that (i) we have access to confidence metrics of the teacher's predictions; and (ii) that often times, when the teacher model's top-1 prediction is inaccurate the true label is within its top-$k$ predictions for some appropriate $k$, to design and estimate a much more refined model for the teacher's noise that we use to inform the design of the student's loss function.

Another related technique for dealing with noisy data is using "robust" loss functions [4, 20, 24, 35, 56] such that they achieve a small risk for new clean examples even under the presence of noise in the training dataset. In Section 4.2 we compare our method with the general framework of [20] for designing robust loss functions and we show that our approach, when applied to the standard cross-entropy loss, consistently outperforms [20] in the setting of distillation with unlabeled examples. That said, we stress that our method is not tied to the cross-entropy loss and, in fact, it often gives better results when combined with more sophisticated loss functions. We demonstrate this in Appendix D.6 where we apply our method in cases where the student loss function comes from the families of losses introduced in [20] and [35].

**Semi-Supervised Learning.** Akin to our setting, in semi-supervised learning (SSL) (see e.g. [55] for a recent survey) the learner is presented with a small labeled dataset $A$ and a typically much larger unlabeled dataset $B$. Unlike to our setting though, there is typically no distinction between the student and teacher: the model of interest generates pseudo-labels on $B$ which are utilized by using appropriate loss functions or preprocessing procedures (e.g. "filtering" or "correcting") — often times in an iterative fashion with the goal of improving the quality of the newly-generated pseudo-labels. It is also worth noting that in many real-world applications of distillation with unlabeled examples either the teacher model is unavailable or it is too expensive to retrain it and create fresh pseudo-labels on the data (e.g., when we request labels from a pretrained large language model). Therefore, SSL approaches that either (i) update the "teacher" model (e.g., [34]), or (ii) require several fresh teacher-generated pseudo-labels (e.g., by requesting teacher-predictions on random data-augmentations or perturbed version of the unlabeled examples of $B$ e.g., [9]) are not applicable in our setting. We implement the recent SSL technique of [48] and show that our method outperforms it in the context of distillation with unlabeled examples. Besides performing on par with state-of-the-art SSL approaches like [9], the method of [48] is free of inherent limitations like using domain-specific data augmentations — which is also an important feature of our approach.

**Learning Halfspaces with Random Classification Noise.** The theoretical study of classification with Random Classification Noise (RCN) was initiated by [5]. For the fundamental class of linear classifiers (halfspaces) the first polynomial time algorithms for the problem where given in [13] and [10]. The iteration proposed in [13] is a "backward loss-adjustment" method [43] for which it is known that resulting optimization landscape is convex (for linear classifiers). In [19] an improved analysis of the method of [13] was given, showing that SGD on this convex loss learns $\gamma$-margin halfspaces with RCN with $\widetilde{O}(1/(\gamma^4 \epsilon^2))$ samples. On the other hand, forward loss-adjustment methods for dealing with RCN are known to result in an inherently non-convex landscape, see [38] and Figure 9). Our theoretical result for SLaM (see Theorem 5.1) is the first convergence result for a "forward loss-adjustment" method and, at the same time, achieves a sample complexity of $O(1/(\gamma^2 \epsilon^2))$ improving over the prior work.

## 3 SLaM: Student-Label Mixing Distillation

In this section, we describe our distillation with unlabeled examples setting and present SLaM. In what follows, we assume that examples are represented by feature-vectors in some space $\mathcal{X}$. We shall denote by $X$ the distribution over examples. We consider multi-class classification with $L$ classes and assume that the ground-truth label of an example $x$ is represented by a one-hot vector in $\mathcal{Y} = \{0, 1\}^L$ given by some unknown function $g(x) : \mathcal{X} \mapsto \mathcal{Y}$. In multi-class classification the learning algorithm typically

optimizes a parametric family of classification models $\mathcal{F} = \{f(\cdot; w) : \mathcal{X} \mapsto \mathbf{R}^L : w \in \mathcal{W}\}$, i.e., for every parameter $w \in \mathcal{W}$, $f(x; w)$ is an $L$-dimensional "score vector", where $f(x; w)_i$ corresponds to the probability that the model assigns to the class $i$ for the example $x$. We shall denote by $\ell(\cdot, \cdot) : \mathbf{R}^L \times \mathbf{R}^L \mapsto \mathbf{R}$ the classification loss function used by the learning algorithm. During training the algorithm considers a set of labeled examples $S = \{(x^{(1)}, g(x^{(1)})), \ldots, (x^{(n)}, g(x^{(n)}))\}$ and optimizes the loss $\ell(\cdot, \cdot)$ over $S$, i.e., solves the problem $\min_{w \in \mathcal{W}} \frac{1}{|S|} \sum_{(x, g(x)) \in S} \ell(g(x), f(x; w))$. For two vectors $v, u \in \mathbf{R}^L$ we denote by $\mathrm{err}(v, u) = \mathbf{1}\{\mathrm{argmax}(v) \neq \mathrm{argmax}(u)\}$ the indicator of the event that the positions of the maximum elements of $v, u$ agree. Similarly, for two classifiers $h(x), f(x) : \mathbf{R}^d \mapsto \mathbf{R}^L$ we can use $\mathrm{err}(h(x), f(x))$ to denote whether their top-1 predictions for the example $x$ agree. Our goal is to train a classifier over the sample $S$ so that its generalization error, i.e., $\mathbf{E}_{x \sim X}[\mathrm{err}(f(x; w), g(x))]$, is small.

**Distillation with Unlabeled Examples.** We assume that we are given a (usually small) dataset $A$ of correctly labeled examples $(x, g(x))$ and a set of unlabeled data $U$. A "teacher" model $y_s(\cdot) : \mathcal{X} \mapsto \mathbf{R}^L$ is first trained on the labeled dataset $A$ and then provides soft-labels for the examples of dataset $U$, i.e., we create a dataset $B = \{(x, y_s(x)) : x \in U\}$ containing examples labeled with the corresponding probability distribution over classes (soft-labels) of the teacher model. We then train a (typically smaller) student model using both the original labeled data $A$ and the teacher-labeled dataset $B$, i.e., $\min_{w \in \mathcal{W}} \frac{1}{|A \cup B|} \sum_{(x, z) \in A \cup B} \ell(z, f(x; w))$. In what follows, we shall call the above training procedure as "vanilla-distillation".

*Remark* 3.1 ("Hard-" vs "Soft-" Distillation). We remark that the process where instead of using the soft-labels provided by the teacher model on the unlabeled dataset U, we use one-hot vectors representing the class with maximum score according to the teacher, is known as hard-distillation. We will denote by $y_s(x)$ the soft-label of the teacher and by $y(x)$ the corresponding hard-label, i.e., $y(x)$ is the one-hot representation of $\mathrm{argmax}\, y_s(x)$. When it is clear from the context we may simply write $y$ instead of $y(x)$.

**Modelling the Teacher as a "Noisy" Label Oracle.** In the distillation setting described in the previous paragraph, it is known [51, 27, 8, 44] that *the teacher model often generates incorrect predictions on the unlabeled examples, impairing the student's performance*. Given any $x \in U$, we model the teacher's prediction $y$ as a random variable. Similarly to [27] we assume that, for every unlabeled datapoint $x \in U$, the provided teacher label $y$ is correct with probability $\alpha(x)$ and incorrect with probability $1 - \alpha(x)$. However, in contrast with [27], our noise model prescribes a non-advsersarial (semi-random) behavior of the teacher when its top-1 prediction is incorrect.

A first step towards more benign noisy teachers is to assume that, conditionally on being wrong, the teacher label is a uniformly random class of the remaining $L - 1$ classes. We remark that this model is already enough to give improvements in datasets with a moderately large number of classes (e.g., up to 100). In particular, it perfectly captures the noisy teacher in binary classification: when the teacher label is different than the ground-truth $g(x)$ then it has to be equal to the "flipped" ground-truth $1 - g(x)$.

We now further refine our model so that it is realistic for datasets with thousands of classes. Even though the top-1 accuracy of the teacher model may not be very high on the unlabeled data $U$, the true label is much more likely to belong in the top-5 or top-10 predictions of the teacher rather than being completely arbitrary. For example, training a ResNet50 network on $10\%$ of ImageNet [49] yields an average top-1 accuracy about $52.78\%$ on the test dataset whereas the top-10 accuracy of the same model is about $83.55\%$. In datasets with a large number of classes, this observation significantly reduces the number of potential correct classes of the examples where the teacher label is incorrect. Motivated by the above, we assume the following structured, semi-random noise model for the teacher, tailored to multi-class settings.

**Definition 3.2** (Noisy Teacher Model). Let $x$ be any example of the unlabeled data $U$ and denote by $g(x)$ its ground-truth label. Let $y_s(x)$ resp. $y(x)$ be the random variable corresponding to the soft resp. hard prediction of the teacher model for the example $x$. We assume that for every $x$ there exist (unknown to the learner) $\alpha(x) \in [0, 1]$ and $k(x) \in \{2, \ldots, L\}$ such that the teacher's top-1 prediction $y$ agrees with the ground-truth $g(x)$ with probability $\alpha(x)$ and, with probability $1 - \alpha(x)$: (i) the

ground-truth belongs in the top-$k(x)$ predictions of the teacher; and (ii) the teacher's (hard)-prediction is a uniformly random *incorrect* class out of the top-$k(x)$ predictions of the teacher soft-label $y_s(x)$ [1].

*Remark* 3.3. We remark that the model of Definition 3.2 captures having a "perfect" teacher model by setting $\alpha(x) = 1$ for all $x$ and also generalizes the binary case described above by taking $k(x) = 2$ for all $x \in X$.

Given the above noise model for the teacher, the problem of improving knowledge-distillation consists of two main tasks: (i) obtaining estimates for accuracy statistics $\alpha(x), k(x)$ for each example $x \in U$; and (ii) using those estimated values to improve the training of the student model so that it is affected less by the mistakes of the teacher on dataset $B$.

**Training Better Students Using $\alpha(x), k(x)$**   We first assume that for every $x$ we have oracle access to the values $\alpha(x), k(x)$ and present our Student-Label Mixing loss function. Instead of using $\alpha(x), k(x)$ to "denoise" the teacher's label, we use them to *add noise to the student's predictions*. To make notation more compact, in what follows, given a vector $z \in \mathbf{R}^L$ we denote by $\mathrm{top}(z; k)$ the vector that has the value 1 in the positions of the of the 1-st up to $k$-th largest elements of $z$ and 0 in all other positions, e.g., $\mathrm{top}((1,2,3);1) = (0,0,1)$ and $\mathrm{top}((-1,1,0,2);3) = (0,1,1,1)$. Assuming that the student-label for some $x \in U$ is $f(x;w)$ we "mix" it (hence the name Student-Label Mixing) using $\alpha(x), k(x)$ to obtain the mixed prediction

$$\mathrm{mix}(f(x;w); \alpha(x), k(x)) = \alpha(x)f(x;w) \ + \ (1 - \alpha(x)) \, \mathrm{top}(y_s(x); k(x)) * \frac{1 - f(x;w)}{k(x) - 1}, \quad (1)$$

where $q * p$ is the element-wise multiplication of the vectors $p, q$. We then train the **mixed** student model, on the "noisy" dataset $B$:

$$\min_{w \in \mathcal{W}} \frac{1}{|A \cup B|} \left( \sum_{(x,z) \in A} \ell(z, f(x;w)) + \sum_{(x,y) \in B} \ell(y, \mathrm{mix}(f(x;w); \alpha(x), k(x))) \right) \quad (2)$$

The main intuition behind the mixing of the student's labels is that *by training the "noisy" student to match the "noisy" teacher label $y$ on dataset $B$, the underlying (non-mixed) student $f(x;w)$ will eventually learn the ground-truth*. In particular, when $\ell(\cdot, \cdot)$ is the Cross-Entropy loss we have that the expected mixed loss conditioned on any $x$ is

$$\mathbf{E}[\ell(y; \mathrm{mix}(f(x;w), a(x), k(x))) \mid x] = \ell(\mathrm{mix}(g(x); \alpha(x), k(x)), \mathrm{mix}(f(x;w); \alpha(x), k(x))),$$

where we used the fact that the cross-entropy is linear in its first argument, and that by the definition of our noise model (Definition 3.2) it holds that $\mathbf{E}[y \mid x] = \mathrm{mix}(g(x); \alpha(x), k(x))$. Therefore, when the student is equal to the ground-truth $f(x;w) = g(x)$, we obtain that the mixed student-model will satisfy $\mathrm{mix}(g(x); \alpha(x), k(x)) = \mathrm{mix}(f(x;w); \alpha(x), k(x))$ for all $x \in X$, and (by Gibb's inequality), we obtain that $g(x)$ is a minimizer of the SLaM loss. We show the following proposition, see Appendix C for the formal statement and proof.

**Proposition 3.4** (SLaM Consistency (Informal))**.** *Let $D$ be the distribution of the teacher-labeled examples of dataset $B$, i.e., we first draw $x \sim X$ and then label it using the noisy teacher of Definition 3.2. Moreover, assume that there exists some parameter $w^* \in \mathcal{W}$ such that the ground-truth $g(x) = f(x; w^*)$. Then $w^*$ is the minimizer of the (population) SLaM objective: $\min_w \mathbf{E}_{(x,y) \sim D}[\mathrm{ce}(y, \mathrm{mix}(f(x;w); \alpha(x), k(x)))]$, where $\mathrm{ce}(\cdot, \cdot)$ is the Cross-Entropy loss.*

**Estimating the Teacher's Accuracy Statistics $\alpha(x), k(x)$ via Isotonic Regression**   We first show how we estimate $\alpha(x)$ for each $x$ of dataset $B$, i.e., the dataset labeled by the teacher model. In [27] the authors empirically observed that $\alpha(x)$ correlates with metrics of teacher's confidence such as the "margin", i.e., the difference between the probabilities assigned in the top-1 class and the second largest class according to the teacher's soft label $y_s$. In particular, the larger the margin is the more likely is that the corresponding teacher label is correct. We exploit this monotonicity by employing isotonic regression on a small validation dataset to learn the mapping from the teacher's margin at an example $x$ to the corresponding teacher's accuracy $\alpha(x)$. For more details, see Appendix B.1.

To perform this regression task we use a small validation dataset $V$ with correct labels that the teacher has not seen during training. For every example $x \in V$ we compute the corresponding

---

[1]Given that the teacher's prediction is incorrect and that the ground-truth belongs in the top-k(x) predictions of the teacher, assumption (ii) describes a uniform distribution on $k(x) - 1$ labels.

soft-teacher label $y_s(x)$ and compute its margin $\mathrm{margin}(x) = \max_1(y_s(x)) - \max_2(y_s(x))$. For every $x \in V$ we also compute the hard-prediction of the teacher and compare it with the ground-truth, i.e., for every $x \in V$ the covariate and responce pair is $(\mathrm{margin}(x), 1 - \mathrm{err}(g(x), y(x)))$. We then use isotonic regression to fit a piecewise constant, increasing function to the data. We remark that isotonic regression can be implemented very efficiently in $O(n \log n)$ time (where $n$ is the size of the validation dataset).

For $k(x)$ we consider two different options: (i) using the same value for all examples (e.g., using $k$ so that the top-k accuracy of teacher is above some threshold on the validation data); and (ii) using a "data-dependent" $k(x)$ that we estimate by solving $L$ (recall that $L$ is the number of classes) isotonic-regression problems (similar to that for estimating $\alpha(x)$ above). We refer to Appendix B.1 for more details.

## 4 Experimental Evaluation

In this section, we present our experimental results. In Section 4.1 we describe our experimental setup and in Section 4.2 we compare the performance of our method with previous approaches on standard benchmarks. In Section D.4 we show that our method can be combined with teacher-uncertainty-based reweighting techniques. Finally, due to space limitations, we provide additional empirical results in the Appendix: in Appendix D.5 we show that SLaM can effectively be used with distillation temperature, and in Appendix D.6 we consider using SLaM with other losses beyond the Cross-Entropy.

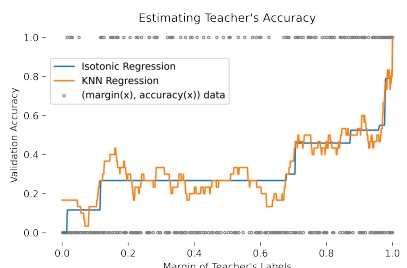

Figure 1: Learning $\alpha(x)$ via isotonic regression. The data were generated by a ResNet 110 teacher trained on 5000 examples of CIFAR-100 and evaluated on a validation dataset $V$ of 500 examples. The regression data $\{(\mathrm{margin}(y_s(x)), 1 - \mathrm{err}(y_s(x), g(x))) : x \in V\}$ are shown in gray (the response is binary $0/1$). By enforcing monotonicity, isotonic regression yields a more stable and robust curve than, for example, the KNN predictor.

### 4.1 The Setup

Here, we describe our procedure for simulating knowledge distillation with unlabeled examples on academic datasets. We start by splitting the training dataset in two parts: dataset A and dataset C. We then train the teacher and student models on dataset A (using the standard cross-entropy loss).[2] Then we perform multiple independent trials where, for each trial, we randomly split dataset C into a small (e.g., 500 examples validation dataset V and an unlabeled training dataset U. For each trial we (i) use the teacher model to label the points on dataset U to obtain the teacher-labeled dataset B (ii) initialize the weights of the student to those of the student model that was pre-trained on dataset A; (iii) train the student model (using each distillation method) on the combined labeled data of A, V (that have true labels) and the data of B (that have teacher labels). We remark here that we include the validation data V during the training of the student to be fair towards methods that do not use a validation dataset. However, while it is important that the teacher has not seen the validation data during training, the performance of no method was affected significantly by including (or excluding) the validation data from the training dataset.

### 4.2 Comparison with Previous Approaches

**The Baselines**   A natural question is whether a more sophisticated distillation method that enforces greater consistency between the teacher and the student, would improve distillation with unlabeled examples: we use the VID method [2] that incorporates the penultimate layer of the student model (after a suitable trainable projection) in the loss. We also compare our method against the weighted distillation method of [27] that reweights the examples of dataset $B$ in order to "correct" the effect of the noisy pseudo-labels provided by the teacher. The Taylor cross-entropy method of [20] is

---

[2]We remark that our method does not require pre-training the student on dataset A, however, since [27] requires pre-training the student, we do the same for all methods that we compare.

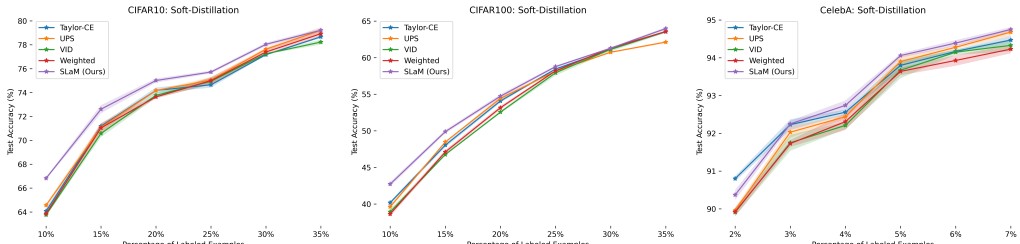

Figure 2: Comparison of distillation methods on CIFAR-10,100 and CelebA. On the horizontal axis we plot the size of Dataset A as a percentage of the whole training dataset. On the vertical axis we plot the accuracy of the trained student-model on the test dataset.

Table 1: Experiments on CIFAR-10 (**soft**-distillation). See Section 4.2 for details.

| Labeled Examples | 5000 | 7500 | 10000 | 12500 | 15000 | 17500 |
|---|---|---|---|---|---|---|
| Teacher | 61.30 | 68.98 | 72.42 | 73.92 | 76.63 | 78.63 |
| Vanilla | $63.53 \pm 0.29$ | $70.39 \pm 0.11$ | $73.23 \pm 0.15$ | $74.29 \pm 0.25$ | $76.64 \pm 0.20$ | $78.63 \pm 0.16$ |
| Taylor-CE [20] | $64.07 \pm 0.26$ | $71.19 \pm 0.17$ | $74.18 \pm 0.25$ | $74.65 \pm 0.24$ | $77.17 \pm 0.04$ | $78.67 \pm 0.13$ |
| UPS [48] | $64.56 \pm 0.13$ | $71.10 \pm 0.34$ | $74.17 \pm 0.06$ | $75.05 \pm 0.24$ | $77.64 \pm 0.12$ | $\mathbf{79.21 \pm 0.27}$ |
| VID [3] | $63.76 \pm 0.13$ | $70.58 \pm 0.17$ | $73.77 \pm 0.40$ | $74.95 \pm 0.21$ | $77.25 \pm 0.16$ | $78.23 \pm 0.09$ |
| Weighted [27] | $63.85 \pm 0.13$ | $71.04 \pm 0.24$ | $73.64 \pm 0.36$ | $75.00 \pm 0.17$ | $77.40 \pm 0.17$ | $78.93 \pm 0.19$ |
| SLaM (Ours) | $\mathbf{66.82 \pm 0.61}$ | $\mathbf{72.61 \pm 0.30}$ | $\mathbf{75.01 \pm 0.25}$ | $\mathbf{75.72 \pm 0.17}$ | $\mathbf{78.04 \pm 0.16}$ | $\mathbf{79.22 \pm 0.11}$ |

a modification of CE that truncates the taylor-series of the CE loss. In [20] it was shown that it offers significant improvements when the labels are corrupted by random classification noise. The fact that the teacher's noise is much closer to random than to adversarial makes this approach a natural baseline. The UPS loss of [48] is a semi-supervised technique that takes into account the variance (uncertainty) of the teacher model on the examples of dataset $B$ in order to transform the soft pseudo-labels provided by the teacher to more "robust" binary vectors and then use a modified binary CE loss. To estimate the uncertainty of the teacher model, we used either dropout with Monte-Carlo estimation or random data-augmentations as suggested in [48]. We remark that, as we discussed in Section 2 and Section 1, strictly speaking, this method is not applicable in our setting because it requires multiple forward passes of the teacher model to estimate its variance but we implement it as it is a relevant approach that aims to improve the pseudo-labels of the teacher.

**CIFAR-{10,100} and CelebA** Here we present our results on CIFAR-{10, 100} [30] and CelebA [22]. CIFAR-10 and CIFAR-100 are image classification datasets with 10 and 100 classes respectively. They contain 60000 labeled images, which are split to a training set of 50000 images, and a test set of 10000 images. From the 50000 images of the train set we use the $10\%, 15\%, 20\%, 25\%, 30\%, 35\%$ (or 5000, 7500, 10000, 12500, 15000, and 17500 examples) as the labeled dataset A where we train the teacher and pre-train the student models. For each size of dataset A, we perform a random split on the remaining training data and use 500 labeled examples as the validation dataset and the remaining examples as the unlabeled dataset U. For the CIFAR-10 experiments, we use a Mobilenet with depth multiplier 2 as the teacher, and a Mobilenet with depth multiplier 1 as the student. For CIFAR-100, we use a ResNet-110 as a teacher, and a ResNet-56 as the student. We compare the methods both on soft- and hard-distillation. For each trial we train the student model for 200 epochs and keep the best test accuracy over all epochs. We perform 3 trials and report the average of each method and the variance of the achieved accuracies over the trials. The results of our experiments for soft-distillation can be found in Table 1 and Table 2. The corresponding plots are given inFigure 2. We include our results for hard-distillation in Appendix D.2.

We consider the male/female binary classification task using the CelebA dataset [22] consisting of a training set of 162770 images and a test set of 19962 images. We use a MobileNet with depth multiplier 2 as the teacher, and a ResNet-11 as the student. As the labeled dataset A we used $2\%, 3\%, 4\%, 5\%, 6\%$ percent (or 3256, 4883, 6510, 8138, 9766, 11394 examples) of the training dataset and split the remaining data in a validation dataset of 500 examples and an unlabeled dataset U. Our results for CelebA can be found in Table 3 (soft-distillation) and in Table 7 (hard-distillation).

Table 2: Experiments on CIFAR-100 (**soft**-distillation). See Section 4.2 for details.

| Labeled Examples | 5000 | 7500 | 10000 | 12500 | 15000 | 17500 |
|---|---|---|---|---|---|---|
| Teacher | 35.97 | 44.65 | 49.62 | 55.68 | 59.19 | 62.05 |
| Vanilla | $37.94 \pm 0.10$ | $46.42 \pm 0.24$ | $52.17 \pm 0.21$ | $57.72 \pm 0.17$ | $60.91 \pm 0.07$ | $63.47 \pm 0.23$ |
| Taylor-CE [20] | $40.18 \pm 0.07$ | $48.05 \pm 0.29$ | $54.08 \pm 0.24$ | $58.45 \pm 0.17$ | $61.13 \pm 0.10$ | $63.54 \pm 0.26$ |
| UPS [48] | $39.62 \pm 0.23$ | $48.48 \pm 0.15$ | $54.43 \pm 0.27$ | $58.17 \pm 0.07$ | $60.74 \pm 0.10$ | $62.13 \pm 0.12$ |
| VID [3] | $38.93 \pm 0.39$ | $46.76 \pm 0.10$ | $52.56 \pm 0.17$ | $57.94 \pm 0.37$ | $61.14 \pm 0.28$ | $63.56 \pm 0.18$ |
| Weighted [27] | $38.63 \pm 0.32$ | $47.11 \pm 0.29$ | $53.16 \pm 0.25$ | $58.20 \pm 0.11$ | $\mathbf{61.29 \pm 0.15}$ | $63.58 \pm 0.07$ |
| SLaM (Ours) | $\mathbf{42.72 \pm 0.30}$ | $\mathbf{49.89 \pm 0.23}$ | $\mathbf{54.73 \pm 0.27}$ | $\mathbf{58.78 \pm 0.15}$ | $\mathbf{61.30 \pm 0.09}$ | $\mathbf{63.98 \pm 0.19}$ |

Table 3: Experiments on CelebA (**soft**-distillation). See Section 4.2 for details.

| Labeled Examples | 2% | 3% | 4% | 5% | 6% | 7% |
|---|---|---|---|---|---|---|
| Teacher | 86.19 | 88.25 | 88.95 | 91.31 | 92.09 | 92.62 |
| Vanilla | $89.96 \pm 0.08$ | $91.55 \pm 0.14$ | $92.16 \pm 0.10$ | $93.42 \pm 0.06$ | $93.98 \pm 0.04$ | $94.29 \pm 0.03$ |
| Taylor-CE [20] | $\mathbf{90.80 \pm 0.07}$ | $\mathbf{92.23 \pm 0.1}$ | $92.56 \pm 0.14$ | $93.80 \pm 0.20$ | $94.17 \pm 0.07$ | $94.47 \pm 0.01$ |
| UPS [48] | $89.96 \pm 0.11$ | $92.03 \pm 0.09$ | $92.44 \pm 0.04$ | $93.9 \pm 0.05$ | $94.28 \pm 0.07$ | $94.68 \pm 0.03$ |
| VID [3] | $89.91 \pm 0.10$ | $91.75 \pm 0.21$ | $92.21 \pm 0.10$ | $93.67 \pm 0.21$ | $94.15 \pm 0.07$ | $94.33 \pm 0.16$ |
| Weighted [27] | $89.92 \pm 0.12$ | $91.73 \pm 0.09$ | $92.31 \pm 0.22$ | $93.64 \pm 0.10$ | $93.93 \pm 0.14$ | $94.23 \pm 0.11$ |
| SLaM (Ours) | $90.37 \pm 0.17$ | $\mathbf{92.25 \pm 0.11}$ | $\mathbf{92.74 \pm 0.17}$ | $\mathbf{94.06 \pm 0.07}$ | $\mathbf{94.39 \pm 0.10}$ | $\mathbf{94.75 \pm 0.08}$ |

The corresponding plots are given in Figure 2. Due to space limitations our results for hard-distillation can be found in Appendix D.2.

Taken together, our comparisons show that SLaM consistently outperforms the baselines, often by a large margin. The reader is referred to Appendix D.1 for additional details.

*Remark* 4.1 (Soft-Distillation and Temperature Scaling). We remark that in the comparisons we performed soft-distillation with temperature set to 1, i.e., for every example we do not scale the corresponding teacher and student logits. In Appendix D.5 we show that our method can readily be used together with temperature scaling to improve the accuracy of the student model.

**ImageNet** Here we present the results on ImageNet [49]. ImageNet is an image classification dataset with 1000 classes consisting of a training set of approximately 1.3 million images, and a test set of 50000 images. From the 1.3 million images of the training set we use the $5\%, 10\%, 15\%, 20\%$ percent (or 64058, 128116, 192174, 256232 examples) as the labeled dataset $A$ where we train the teacher and pre-train the student models. For each size of dataset $A$, we perform a random split on the remaining training data and use 10000 labeled examples as the validation dataset and the remaining examples as the unlabeled dataset $U$. We use a ResNet-50 as the teacher, and a ResNet-18 as the student. We compare the methods on soft-distillation. For each trial, we train the student model for 100 epochs

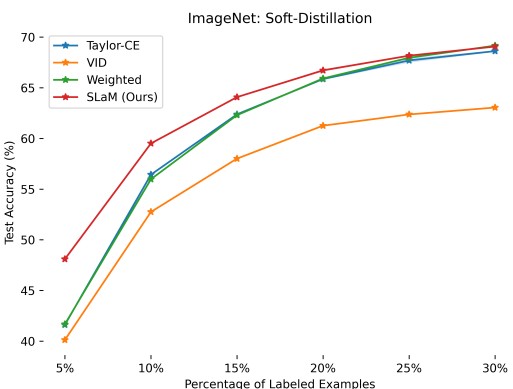

Figure 3: Comparison of distillation methods on ImageNet. On the horizontal axis we plot the size of Dataset A as a percentage of the whole training dataset. On the vertical axis we plot the accuracy of the trained student-model on the test dataset.

and keep the best test accuracy over all epochs. We perform 4 trials and report the average of each method and the variance of the achieved accuracies over the trials. Our results for ImageNet can be found in Table 4. We remark that we do not include the results of the UPS method in Table 4 because it did not improve over the accuracy achieved after pre-training the student model on dataset $A$. The reader is referred to Appendix D.1 for additional details.

Table 4: Experiments on ImageNet (**soft**-distillation). See Section 4.2 for details.

| Labeled Examples | 5% | 10% | 15% | 20% | 25% | 30% |
|---|---|---|---|---|---|---|
| Teacher | 39.48 | 52.96 | 59.64 | 63.62 | 66.00 | 67.85 |
| Vanilla | $41.67 \pm 0.05$ | $55.9 \pm 0.06$ | $62.3 \pm 0.09$ | $65.91 \pm 0.05$ | $67.98 \pm 0.07$ | $69.12 \pm 0.08$ |
| Taylor-CE [20] | $41.61 \pm 0.06$ | $56.43 \pm 0.06$ | $62.38 \pm 0.11$ | $65.86 \pm 0.08$ | $67.70 \pm 0.22$ | $68.62 \pm 0.07$ |
| VID [3] | $40.12 \pm 0.04$ | $52.75 \pm 0.04$ | $58.01 \pm 0.03$ | $61.21 \pm 0.06$ | $62.37 \pm 0.06$ | $63.05 \pm 0.07$ |
| Weighted [27] | $41.67 \pm 0.04$ | $55.96 \pm 0.07$ | $62.29 \pm 0.08$ | $65.91 \pm 0.05$ | $67.96 \pm 0.06$ | $\mathbf{69.16 \pm 0.08}$ |
| SLaM (Ours) | $\mathbf{48.1 \pm 0.05}$ | $\mathbf{59.51 \pm 0.07}$ | $\mathbf{64.08 \pm 0.06}$ | $\mathbf{66.72 \pm 0.11}$ | $\mathbf{68.17 \pm 0.07}$ | $69.07 \pm 0.05$ |

**Large Movies Reviews Dataset** Here we present results on the Large Movies Reviews Dataset [39]. This is a dataset for binary sentiment classification containing 25000 movie reviews for training and 25000 for testing. We use an ALBERT-large model [32] as a teacher, and an ALBERT-base model as a student. We use $2\%, 4\%, 8\%, 40\%$ percent (or 500, 1000, 2000, 10000 examples) from the training dataset and split the remaining data in a validation dataset of 500 examples and an unlabeled dataset $U$. Our results and more experimental details can be found in Appendix D.3.

**Performance Gains of SLaM as a Function of The Number of Labeled Examples** In our experiments, the fraction of examples we consider "labeled" controls two things at the same time: (i) the accuracy of the teacher model — as the teacher is trained on the labeled examples available; and (ii) the number of unlabeled examples the teacher model provides pseudo-labels for. The more inaccurate the teacher model is, the better the improvements provided by our method. (Given a "perfect" teacher that never generates incorrect pseudo-labels for the unlabeled examples, our method is mathematically equivalent to the "vanilla" approach (see the mixing operation in Equation (1)). Therefore, the smaller the number of labeled examples available, the bigger the performance gains of SLaM as (i) the teacher will be less accurate; and (ii) it has to generate labels for more unlabeled examples (and therefore the absolute number of inaccurate predictions that SLaM "corrects" increases statistically). It is worth emphasizing that the main reason behind the enormous success of distillation is exactly that the teacher network can blow up the size of the student's training dataset: in practice, the ratio of labeled examples to unlabeled examples is typically (much) less than 1%.

## 5 Distilling Linear Models and Learning Noisy Halfspaces

In this section we show that, when the dataset is separable by a halfspace, i.e., for every example $x$, the ground-truth is $g(x) = (\mathbf{1}\{w^* \cdot x > 0\}, \mathbf{1}\{w^* \cdot x \leq 0\})$ for some unknown weight vector $w^*$, then using SLaM with a linear model as the student will recover the ground truth classifier. We make the standard assumption that the ground-truth halfspace has $\gamma$-margin, i.e., that $\|w^*\|_2 = 1$ and that it holds $|w^* \cdot x| \geq \gamma$ for all examples $x$. For a fixed example $x$, the observed noisy teacher-label $y$ satisfies Definition 3.2, i.e., $y = g(x)$ w.p. $\alpha(x)$ and $y = 1 - g(x)$ w.p. $1 - \alpha(x)$ (since $k = 2$ for binary classification). Our approach consists of using the standard cross-entropy loss $\mathrm{ce}(p, q)$ and training a student-model consisting of a linear layer plus a soft-max activation, i.e., $f(x; w) = \left( \frac{1}{1+e^{-w \cdot x}}, \frac{e^{-w \cdot x}}{1+e^{-w \cdot x}} \right)$.

**Theorem 5.1** (SLaM Convergence). *Let $X$ be a distribution on $\mathbf{R}^d$ and $g(x)$ be the ground-truth halfspace with normal vector $w^* \in \mathbf{R}^d$. Let $D$ be the distribution over (noisy) teacher-labeled examples $(x, y)$ whose $x$-marginal is $X$. Assume that there exist $\beta, \gamma > 0$ such that for all examples $x$ in the support of $X$ it holds that $|w^* \cdot x| \geq \gamma$ and $|1/2 - \alpha(x)| \geq \beta$. Let $\epsilon > 0$. After $T = O(1/(\beta^2 \gamma^2 \epsilon^2))$ SGD iterations on the SLaM objective (see Algorithm 3), with probability at least $99\%$, there exists an iteration $t \leq T$ where $\mathbf{P}_{x \sim X}[\mathrm{err}(f(x; w^{(t)}), g(x))] \leq \epsilon$.*

*Remark* 5.2 (Learning Halfspaces with RCN). The problem of learning halfspaces with Random Classification Noise (RCN) can be modeled as having a teacher with constant accuracy probability, i.e., $\alpha(x) = \alpha > 1/2$ for all $x$. As a corollary of Theorem 5.1 we obtain an efficient learning algorithm for $\gamma$-margin halfspaces under RCN achieving a sample complexity of $O(1/(\gamma^2 \epsilon^2))$. Prior to our work, the best known sample complexity for provably learning halfspaces with RCN was $\widetilde{O}(1/(\gamma^4 \epsilon^2))$ [19] where the "backward loss-adjustment" of [13] was used.

# 6    Conclusion, Limitations, and Broader Impact

In this work we propose SLaM, a novel and principled method for improving distillation with unlabeled examples. We empirically show that SLaM consistently outperforms the baselines, often by a large margin. We also showed that SLaM can be used with and improve (i) knowledge distillation with temperature scaling; (ii) loss functions beyond the standard Cross-Entropy loss; and (iii) confidence-based weighting schemes that down-weight examples where the teacher model is not very confident. Apart from extensive experimental evaluation, we provide strong theoretical guarantees establishing the consistency and optimality of SLaM. As a byproduct of our theoretical analysis, we obtain a new iteration for learning $\gamma$-margin halfspaces with RCN that improves the best known sample complexity for this problem.

A limitation of SLaM is that it does not necessarily improve over vanilla distillation when the teacher model makes only a few mistakes (this is to be expected as our method is designed for the case where the teacher-model is imperfect).

Knowledge-distillation is a very popular deep learning method, and therefore, potentially malicious usage of our work is an important societal issue, as deep learning has far-reaching applications from NLP to Robotics and Self-Driving cars.

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

# A   Notation

For two vectors $p, q \in \mathbf{R}^d$ we denote by $p \cdot q = \sum_{i=1}^{d} p_i q_i$ their inner product. We use $p * q$ to denote their element-wise product, i.e., $(p * q)_i = p_i q_i$. We use the notation $\max_i p$ to denote the $i$-th largest element of the vector $p$. We use $\mathrm{margin}(p)$ to denote the difference between the top-2 elements of $p$, i.e., $\mathrm{margin}(p) = \max_1 p - \max_2 p$. Moreover, we use $\mathrm{margin}_k(p)$ to denote the top-k margin, i.e., $\mathrm{margin}_k(p) = \sum_{i=1}^{k} \max_i p - \max_{k+1} p$. Given a function $f(w) : \mathbf{R}^d \mapsto \mathbf{R}$ we denote by $\partial_w f(w)$ the gradient of $f$ with respect to the parameter $w$.

# B   Detailed Description of SLaM

## B.1   Estimating the Teacher's Accuracy Parameters: $\alpha(x), k(x)$

**Estimating the Teacher's Accuracy $\alpha(x)$ via Isotonic Regression**   We now turn our attention to the problem of estimating $\alpha(x)$ for each $x$ of dataset $B$, i.e., the dataset labeled by the teacher model. In [27] the authors empirically observed that $\alpha(x)$ correlates with metrics of teacher's confidence such as the "margin", i.e., the difference between the probabilities assigned in the top-1 class and the second largest class according to the teacher's soft label $y_s$. In particular, the larger the margin is the more likely is that the corresponding teacher label is correct. We exploit (and enforce) this monotonicity by employing isotonic regression on a small validation dataset to learn the mapping from the teacher's margin at an example $x$ to the corresponding teacher's accuracy $\alpha(x)$.

To perform this regression task we use a small validation dataset $V$ with correct labels that the teacher has not seen during training. For every example $x \in V$ we compute the corresponding soft-teacher label $y_s(x)$ and compute its margin $\mathrm{margin}(x) = \max_1(y_s(x)) - \max_2(y_s(x))$. For every $x \in V$ we also compute the hard-prediction of the teacher and compare it with the ground-truth, i.e., for every $x$ the covariate and responce pair is $(\mathrm{margin}(x), 1 - \mathrm{err}(g(x), y(x)))$. We then use isotonic regression to fit a piecewise constant, increasing function to the data. Sorting the regression data $\{(\mathrm{margin}(x), 1 - \mathrm{err}(g(x), y(x))) x \in V\}$ by increasing margin to obtain a list $(c^{(1)}, \ldots, r^{(1)}), \ldots, (c^{(m)}, r^{(m)})$, isotonic regression solves the following task

$$\min_{\hat{r}^{(1)}, \ldots, \hat{r}^{(m)}} \sum_{i=1}^{m} (r^{(i)} - \hat{r}^{(i)})^2$$

$$\text{subject to} \quad \mathrm{lb} \leq \hat{r}^{(i)} \leq \hat{r}^{(i+1)} \leq 1,$$

where the parameter $\mathrm{lb}$ is a lower bound on the values $\hat{r}^{(i)}$ and is a hyper-parameter that we tune. On the other hand, the upper bound for the values can be set to $1$ since we know that the true value $\alpha(x)$ is at most $1$ for every $x$ (since it corresponds to the probability that the teacher-label is correct). After we compute the values $\hat{r}^{(1)}, \ldots, \hat{r}^{(m)}$ for any given $c \in [0, 1]$ the output of the regressor is the value of $\hat{r}^{(i)}$ corresponding to the smallest $c^{(i)}$ that is larger-than or equal to $c$. This is going to be our estimate for $\alpha(x)$. We remark that finding the values $r^{(i)}$ can be done efficiently in $O(n)$ time after sorting the data (which has a runtime of $O(n \log n)$) so the whole isotonic regression task can be done very efficiently.

**Estimating $k(x)$.**   We now describe our process for estimating the values of $\alpha(x)$ and $k(x)$ for every example of dataset $B$. Similarly to the binary classification setting, we estimate the accuracy probability $\alpha(x)$ using isotonic regression on a small validation dataset. The value of $k(x)$ can be set to be equal to a fixed value of $k$ for all data, so that the top-k accuracy of the teacher on the validation data is reasonable (say above 60%). For example, in our ImageNet experiments, we used $k = 5$. We also provide a data-dependent method to find different values $k(x)$ for every example $x$. To do this we adapt the method for estimating the top-1 accuracy $\alpha(x)$ of the teacher from the validation dataset. For every value of $k = 2, \ldots, L - 1$ we compute the top-k margin of the teacher's predictions on the validation data which is equal to the sum of the top-k probabilities of the teacher soft-label minus the probability assigned to the $k + 1$-th class, i.e.,

$$\mathrm{margin}_k(y_s(x)) = \left( \sum_{i=1}^{k} \max_i y_s(x) \right) - \max_{k+1} y_s(x).$$

Using the top-k margin as the covariate and the top-k accuracy as the response we solve the corresponding regression task using isotonic regression to obtain the value $\alpha_k(x)$ representing the probability that the true label belongs in the top-k predictions of the teacher soft-label. For some threshold, say 90%, for every $x$ we set $k(x)$ to be the smallest value of $k$ so that $\alpha_k(x) \geq 90\%$. We empirically observed that using larger thresholds for the top-k accuracy (e.g., 90% or 95%), is better. We remark that while using the top-k margin as the covariate in the regression task is reasonable, our method can be used with other "uncertainty metrics" of the teacher's soft-labels, e.g., the entropy of the distribution of $y_s(x)$ after grouping together the top-k elements. The higher this entropy metric is the more likely that the top-k accuracy probability $\alpha(x)_k$ of the teacher is low.

## B.2 SLaM for Distillation with Unlabeled Examples: Pseudocode

In this section we present pseudo-code describing the distillation with unlabeled examples setting and the SLaM method, Algorithm 1.

*Remark* B.1. We remark that in our experiments, we observed that not normalizing the mixing operation with $k(x) - 1$ resulted in better results overall. Therefore, the mixing operation used in our experimental evaluation of SLaM is $\mathrm{mix}(f(x; w); \alpha(x), k(x)) = \alpha(x)f(x; w) + (1 - \alpha(x))(1 - f(x; w)) * \mathrm{top}(y_s(x); k(x))$. For more details we refer the reader to the code provided in the supplementary material.

---

**Algorithm 1** Student Label Mixing (SLaM) Distillation

**Input:** Labeled Dataset A, Labeled Validation dataset V, Unlabeled Dataset U
**Output:** A trained Student model $f(x; w)$

Train Teacher model on Labeled Dataset A
Pre-train Student model on Labeled Dataset A

*# Label examples of Dataset U using the Teacher*
$B \leftarrow \emptyset$
**for** each $x \in U$ **do**
  Add $(x, y_s(x))$ to $B$      *# For hard-distillation use $y(x)$*
**end for**

*# Learn Teacher Accuracy Statistics $\alpha(x), k(x)$ Algorithm 2*
$\hat{\alpha}(x), \hat{k}(x) \leftarrow \mathrm{LearnAccuracyStatistics}(y(\cdot), V, B)$
Train student $f(x; w)$ using the SLaM loss:

$$\sum_{(x,y)\in A\cup V} \ell(y, f(x; w)) + \sum_{(x,y)\in B} \ell(y, \mathrm{mix}(f(x; w); \hat{a}(x), \hat{k}(x)))$$

---

## C    SLaM Consistency

In the following proposition we show that any minimizer of the SLaM loss over the noisy teacher-data must agree with the ground-truth for all $x$ (that have positive density). To keep the presentation simple and avoid measurability issues (e.g., considering measure zero sets under $X$) in the following we will assume that the example distribution $X$ is supported on a finite set. We remark that one can easily adapt the proof to hold for any distribution $X$ (but the result will hold after excluding measure-zero sets under $X$).

**Proposition C.1** (SLaM Consistency). *Let $D$ be the distribution of the teacher-labeled examples of dataset B, i.e., we first draw $x \sim X$ and then label it using the noisy teacher of Definition 3.2. Moreover, assume that there exists some parameter $w^* \in \mathcal{W}$ such that the ground-truth $g(x) = f(x; w^*)$. Denote by $\mathcal{L}^{SLaM}(w) = \mathbf{E}_{(x,y)\sim D}[\ell(y, \mathrm{mix}(f(x; w); \alpha(x), k(x))].$ the SLaM objective. The following hold true.*

*1. $w^*$ minimizes the SLaM objective.*

**Algorithm 2** Estimating Teacher's Accuracy Statistics $\alpha(x), k(x)$

---

**Input:** (Noisy) Teacher Model $y_s(x)$, Labeled Validation dataset V,
Isotonic-Regression lower-bound $\text{lb} \in [0, 1]$, and top-k accuracy threshold $t \in [0, 1]$.
**Output:** Estimates $\hat{\alpha}(x), \hat{k}(x)$ of the actual $\alpha(x), k(x)$.

Create Soft-labels for the Validation dataset using the teacher model $\{y_s(x) : x \in V\}$.
**for** $j = 1$ to $L - 1$ **do**
    *# Map $y_s(x)$ to top-j margin and accuracy pairs on the Validation V*

$$C \leftarrow \left\{ \left( \sum_{r=1}^{j} \max_r y_s(x) - \max_{j+1} y_s(x), \ 1 - \text{err}(y_s(x), z) \right) : (x, z) \in V \right\}.$$

    Set $\hat{\alpha}_j(x)$ to be the output of Isotonic-Regression with lower-bound $\text{lb}$ on the (covariate, responce) pairs in $C$.    *# See Appendix B.1*
**end for**
$\hat{a}(x) \leftarrow \hat{a}_1(x)$
$\hat{a}_L(x) \leftarrow 1$    *# The top-L accuracy is always (trivially) equal to 1*
Given example $x$ for some threshold $t$ set $\hat{k}(x)$ to be the smallest integer $r \in \{1, \ldots, L\}$ so that
$a_r(x) \geq t$.

---

    2. *Assuming further that for all $x$ it holds that $\alpha(x)k(x) \neq 1$, we have that* any *minimizer $w$ of the SLaM objective satisfies: $f(x; w) = g(x)$ for all $x$.*

*Proof.* Fix any example $x \in X$. By Definition 3.2 we have that the corresponding teacher label $y$ is correct with probability $\alpha(x)$ and a uniformly random incorrect label out of the top-k labels according to the teacher soft-label $y_s(x)$. Recall for an $L$-dimension score vector $p$, by $\text{top}(p; k) \in \{0, 1\}^L$ we denote the vector that has 1 on the positions of the top-k elements of $p$, e.g., $\text{top}((1, 2, 3, 4, 5); 2) = (0, 0, 0, 1, 1)$. Conditional on $x$, the corresponding expected noisy teacher label is

$$\mathbf{E}[y \mid x] = \mathbf{P}[y = g(x) \mid x]g(x) + \mathbf{P}[y \neq g(x)]\,\mathbf{E}[y \mid x, y \neq g(x)]$$
$$= \alpha(x)g(x) + (1 - \alpha(x))\,\mathbf{E}[y \mid y \neq g(x), x].$$

We know that the expected teacher label conditional on it being wrong $\mathbf{E}[y \mid y \neq g(x), x]$ is a uniformly random incorrect label from the top-k labels of the corresponding teacher soft-label $y_s(x)$. Assume first that $k = L$, since the ground-truth is represented by a one-hot vector, the distribution of uniformly random incorrect labels conditional on $x$ can be written as $(1 - g(x))/(L - 1)$. For example, if the ground-truth label is $g(x) = (1, 0, 0, 0, 0)$ then a uniformly random incorrect label has probability distribution $(0, 1/4, 1/4, 1/4, 1/4)$. Assume now that $k(x) = 3$ and $\text{top}(y_s(x); 3) = (1, 1, 1, 0, 0)$. Then the distribution of the (incorrect) teacher label becomes $(0, 1/2, 1/2, 0, 0)$. Using $*$ to denote element-wise multiplication of two vectors, we have

$$\mathbf{E}[y \mid x, y \neq g(x)] = \frac{1 - g(x)}{k(x) - 1} * \text{top}(y_s(x); k(x))$$

Therefore, we obtain

$$\mathbf{E}[y \mid x] = \alpha(x)g(x) + (1 - \alpha(x))\frac{1 - g(x)}{k(x) - 1} * \text{top}(y_s(x); k(x)) = \text{mix}(g(x); \alpha(x), k(x)).$$

Therefore, by using the fact that Cross-Entropy is linear in its first argument, we obtain that the expected SLaM loss on some example $x$ is

$$\mathbf{E}[\text{ce}(y, \text{mix}(f(x; w); \alpha(x), k(x))) \mid x] = \text{ce}(\mathbf{E}[y \mid x], \text{mix}(f(x; w); \alpha(x), k(x)))$$
$$= \text{ce}(\text{mix}(g(x; w); \alpha(x), k(x)), \text{mix}(f(x; w); \alpha(x), k(x))).$$

We first have to show that there exist some parameter $w \in \mathcal{W}$ that matches the (expected) observed labels $\mathbf{E}[y \mid x]$. Observe first that by using the realizability assumption, i.e.,that there exists $w^*$ so that $f(x; w^*) = g(x)$ we obtain that, for every $x$, it holds $\text{mix}(g(x); \alpha(x), k(x)) = \text{mix}(f(x; w^*); \alpha(x), k(x))$. In fact, by Gibb's inequality (convexity of Cross-Entropy) we have that $w^*$ is a (global) minimizer of the SLaM objective.

We next show that *any (global) minimizer* of the SLaM objective must agree with the ground-truth for every $x$. Since we have shown that $w^*$ is able to match the (expected) labels $\mathbf{E}[y \mid x]$ any other minimizer $w$ must also satisfy $\mathrm{mix}(g(x); \alpha(x), k(x)) = \mathrm{mix}(f(x; w); \alpha(x), k(x))$. Assume without loss of generality that $g_0 = 1$, i.e., the ground-truth label is 0. We observe that by using that $\mathrm{mix}(g(x; w); \alpha(x), k(x)) = \alpha(x)g(x) + (1 - \alpha(x))\frac{1-g(x)}{k(x)-1} * \mathrm{top}(y_s(x); k(x))$ and the fact that the ground-truth belongs in the top-$k(x)$ of the teacher's predictions conditional that the teacher's top-1 prediction is incorrect (thus $\mathrm{top}(y_s(x))_0 = 1$), we obtain that

$$\alpha(x)g_0(x) + (1-\alpha(x))(1-g_0(x))/(1-k(x)) = \alpha(x)f(x; w)_0 + (1-\alpha(x))(1-f(x; w)_0)/(k(x)-1).$$

Using the fact that $g_0 = 1$ we can simplify the above expression to

$$(1 - f(x; w)_0)\left(\alpha(x) - \frac{1 - \alpha(x)}{k(x) - 1}\right) = 0.$$

Using the assumption that $a(x)k(x) \neq 1$ we obtain that the term $\left(\alpha(x) - \frac{1-\alpha(x)}{k(x)-1}\right)$ is not vanishing and therefore it must hold that $f(x; w)_0 = 1 = g_0$, i.e., the student model must be equal to the ground-truth.

$\square$

## D   Extended Experimental Evaluation

We implemented all algorithms in Python and used the TensorFlow deep learning library [1]. We ran our experiments on 64 Cloud TPU v4s each with two cores.

### D.1   Implementation Details: Vision Datasets

Here we present the implementation details for the vision datasets we considered.

*Remark* D.1.  We note that in all our experiments, "VID" corresponds to the implementation of the loss described in equation (2), (4) and (6) of [2] (which requires appropriately modifying the student model so that we have access to its embedding layer).

**Experiments on CIFAR-{10/100} and CelebA**    For the experiments on CIFAR-10/100 and CelebA we use the Adam optimizer with initial learning rate $\mathrm{lr} = 0.001$. We then proceed according to the following learning rate schedule (see, e.g., [25]):

$$\mathrm{lr} \leftarrow \begin{cases} \mathrm{lr} \cdot 0.5 \cdot 10^{-3}, & \text{if } \#\mathrm{epochs} > 180 \\ \mathrm{lr} \cdot 10^{-3}, & \text{if } \#\mathrm{epochs} > 160 \\ \mathrm{lr} \cdot 10^{-2}, & \text{if } \#\mathrm{epochs} > 120 \\ \mathrm{lr} \cdot 10^{-1}, & \text{if } \#\mathrm{epochs} > 80 \end{cases}$$

Finally, we use data-augmentation. In particular, we use random horizontal flipping and random width and height translations with width and height factor, respectively, equal to $0.1$.

The hyperparameters of each method are optimized as follows. For SLaM we always use $0.5$ as the lower bound for isotonic regression (i.e., the parameter $\mathrm{lb}$ in Algorithm 2). As CelebA is a binary classification benchmark $k(x)$ is naturally set to 2 for all examples. For CIFAR-10/10 we used the data-dependent method for estimating $k(x)$ (see Algorithm 2) with threshold parameter $t = 0.9$. For weighted distillation we do a grid search over updating the weights every $\{1, 25, 50, 100, 200\}$ epochs and we report the best average accuracy achieved. Finally, for VID we search over $\{0.001, 0.1, 0.2, 0.5, 0.8, 1.0, 2.0, 10.0, 50.0, 100.0\}$ for the coefficient of the VID-related term of the loss function, and for the Taylor cross-entropy method we optimize its hyperparameter over $\{1.0, 2.0, 3.0, 4.0, 5.0, 6.0\}$.

**Experiments on ImageNet**    For the ImageNet experiments we use SGD with momentum $0.9$ as the optimizer. For data-augmentation we use random horizontal flipping and random cropping. Finally, the learning rate schedule is as follows. For the first 5 epochs the learning rate $\mathrm{lr}$ is increased from

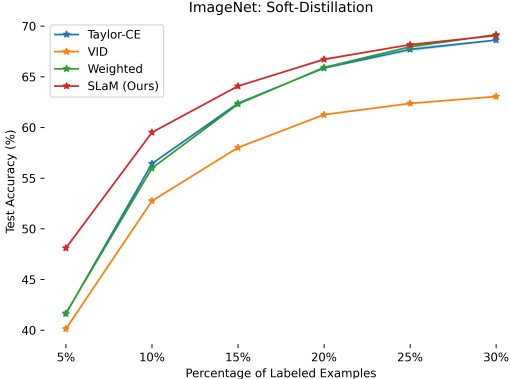

Figure 4: Comparison of distillation methods on ImageNet. On the horizontal axis we plot the size of Dataset A as a percentage of the whole training dataset. On the vertical axis we plot the accuracy of the trained student-model on the test dataset.

Table 5: Experiments on CIFAR-10 (**hard**-distillation). See Section 4.2 for details.

| Labeled Examples | 5000 | 7500 | 10000 | 12500 | 15000 | 17500 |
|---|---|---|---|---|---|---|
| Teacher | 61.30 | 68.98 | 72.42 | 73.92 | 76.63 | 78.63 |
| Vanilla | $62.26 \pm 0.45$ | $69.07 \pm 0.11$ | $72.09 \pm 0.11$ | $73.43 \pm 0.16$ | $75.93 \pm 0.25$ | $77.43 \pm 0.15$ |
| Taylor-CE [20] | $63.14 \pm 0.07$ | $69.98 \pm 0.11$ | $72.72 \pm 0.36$ | $73.77 \pm 0.28$ | $76.26 \pm 0.29$ | $77.88 \pm 0.20$ |
| UPS [48] | $64.27 \pm 0.08$ | $70.93 \pm 0.26$ | $73.78 \pm 0.16$ | $74.66 \pm 0.29$ | $77.38 \pm 0.37$ | $78.95 \pm 0.08$ |
| VID [3] | $61.95 \pm 0.22$ | $66.91 \pm 0.21$ | $69.59 \pm 0.24$ | $72.16 \pm 0.47$ | $74.83 \pm 0.11$ | $75.55 \pm 0.21$ |
| Weighted [27] | $63.22 \pm 0.45$ | $71.04 \pm 0.26$ | $72.84 \pm 0.12$ | $74.20 \pm 0.16$ | $76.56 \pm 0.24$ | $78.23 \pm 0.15$ |
| SLaM (Ours) | $\mathbf{66.40 \pm 0.31}$ | $\mathbf{72.44 \pm 0.17}$ | $\mathbf{74.77 \pm 0.13}$ | $\mathbf{75.64 \pm 0.19}$ | $\mathbf{77.99 \pm 0.36}$ | $\mathbf{79.26 \pm 0.26}$ |

$0.0$ to $0.1$ linearly. After that, the learning rate changes as follows:

$$\text{lr} = \begin{cases} 0.01, & \text{if } \#\text{epochs} > 30 \\ 0.001, & \text{if } \#\text{epochs} > 60 \\ 0.0001, & \text{if } \#\text{epochs} > 80 \,. \end{cases}$$

The hyperparameters of each method are optimized as follows. For SLaM we do a hyperparameter search over $\{0.55, 0.60, 0.65, 0.70\}$ for the lower bound for isotonic regression, and we keep the best performing value for each potential size of dataset $A$. We used the fixed value $5$ for $k(x)$, as the top-5 accuracy of the teacher model was satisfactory (much higher than its top-1 accuracy) on the validation dataset. For Taylor-CE we did a hyper-parameter search for the Taylor series truncation values in $\{1, 2, 3, 4, 5, 6, 10, 20, 50, 80, 100\}$. For weighted distillation we compute the weights in a one-shot fashion using the pre-trained student (as in the ImageNet experiments in [27]). For VID we search over $\{0.1, 0.3, 0.5\}$ for the coefficient of the VID-related term of the loss function.

### D.2   Hard-Distillation

Here we present results on hard-distillation. The hyper-parameters of all methods are chosen the same way as in our soft-distillation experiments, see Appendix D.1. Tables 5, 6 and 7 contain our results on CIFAR-10, CIFAR-100 and CelebA, respectively. We observe that in almost all cases, SLaM consistently outperforms the other baselines. Moreover, for CIFAR-10 and CIFAR-100 hard-distillation performs worse than soft-distillation (as it is typical the case) but in CelebA hard-distillation seems to be performing on par with (sometimes even outperforming) soft-distillation. A plausible explanation for the latter outcome is that in our CelebA experiments the teacher and student have different architectures (MobileNet and ResNet, respectively) so that soft-labels from the teacher are not so informative for the student. (This is also a binary classification task where the information passed from the teacher to the student through its soft-labels is limited.)

Table 6: Experiments on CIFAR-100 (**hard**-distillation). See Section 4.2 for details.

| Labeled Examples | 5000 | 7500 | 10000 | 12500 | 15000 | 17500 |
|---|---|---|---|---|---|---|
| Teacher | 35.97 | 44.65 | 49.62 | 55.68 | 59.19 | 62.05 |
| Vanilla | $36.36 \pm 0.04$ | $44.15 \pm 0.10$ | $50.22 \pm 0.07$ | $55.55 \pm 0.24$ | $58.85 \pm 0.1$ | $61.43 \pm 0.19$ |
| Taylor-CE [20] | $39.12 \pm 0.14$ | $46.87 \pm 0.10$ | $52.64 \pm 0.22$ | $57.19 \pm 0.28$ | $59.95 \pm 0.11$ | $62.36 \pm 0.21$ |
| UPS [48] | $39.49 \pm 0.13$ | $48.36 \pm 0.44$ | $53.95 \pm 0.10$ | $57.95 \pm 0.10$ | $60.59 \pm 0.29$ | $62.09 \pm 0.28$ |
| VID [3] | $37.19 \pm 0.09$ | $44.67 \pm 0.16$ | $50.63 \pm 0.35$ | $54.78 \pm 0.07$ | $59.27 \pm 0.14$ | $62.01 \pm 0.05$ |
| Weighted [27] | $38.04 \pm 0.29$ | $46.45 \pm 0.22$ | $52.33 \pm 0.18$ | $57.43 \pm 0.13$ | $60.81 \pm 0.09$ | $63.02 \pm 0.06$ |
| SLaM (Ours) | $\mathbf{42.01 \pm 0.29}$ | $\mathbf{49.08 \pm 0.14}$ | $\mathbf{54.49 \pm 0.17}$ | $\mathbf{58.53 \pm 0.04}$ | $\mathbf{61.12 \pm 0.15}$ | $\mathbf{63.21 \pm 0.18}$ |

Table 7: Experiments on CelebA (**hard**-distillation). See Section 4.2 for details.

| Labeled Examples | 2% | 3% | 4% | 5% | 6% | 7% |
|---|---|---|---|---|---|---|
| Teacher | 86.19 | 88.25 | 88.95 | 91.31 | 92.09 | 92.62 |
| Vanilla | $89.73 \pm 0.08$ | $91.61 \pm 0.09$ | $92.05 \pm 0.11$ | $93.41 \pm 0.13$ | $94.02 \pm 0.15$ | $94.05 \pm 0.04$ |
| Taylor-CE [20] | $\mathbf{90.62 \pm 0.05}$ | $92.19 \pm 0.02$ | $92.66 \pm 0.11$ | $93.60 \pm 0.14$ | $94.00 \pm 0.04$ | $94.38 \pm 0.10$ |
| UPS [48] | $89.35 \pm 0.04$ | $91.30 \pm 0.04$ | $91.95 \pm 0.12$ | $93.18 \pm 0.07$ | $93.71 \pm 0.04$ | $94.18 \pm 0.03$ |
| VID [3] | $89.92 \pm 0.21$ | $91.60 \pm 0.11$ | $92.20 \pm 0.12$ | $93.51 \pm 0.15$ | $94.08 \pm 0.15$ | $94.27 \pm 0.10$ |
| Weighted [27] | $90.06 \pm 0.06$ | $91.97 \pm 0.13$ | $92.45 \pm 0.10$ | $93.60 \pm 0.07$ | $93.94 \pm 0.12$ | $94.25 \pm 0.16$ |
| SLaM (Ours) | $90.43 \pm 0.05$ | $\mathbf{92.25 \pm 0.11}$ | $\mathbf{92.71 \pm 0.08}$ | $\mathbf{93.96 \pm 0.17}$ | $\mathbf{94.39 \pm 0.21}$ | $\mathbf{94.52 \pm 0.12}$ |

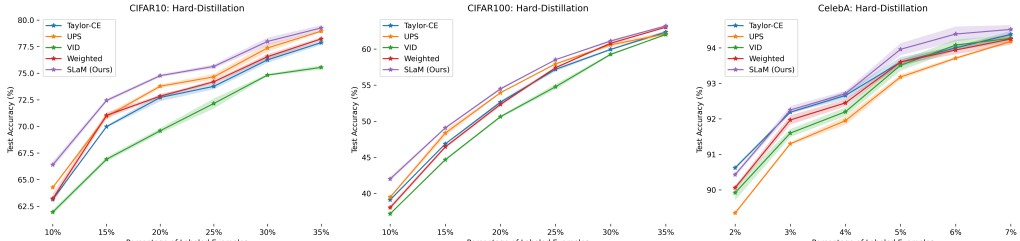

Figure 5: Comparison of distillation methods on CIFAR-10,100 and CelebA. On the horizontal axis we plot the size of Dataset A as a percentage of the whole training dataset. On the vertical axis we plot the accuracy of the trained student-model on the test dataset.

### D.3 Large Movies Reviews Dataset Results

Here we present the results and the implementation details regarding the experiments on the Large Movies Reviews dataset. Recall that we use an ALBERT-large model as a teacher, and an ALBERT-base model as a student. We also use $2\%, 4\%, 8\%, 40\%$ percent (or 500, 1000, 2000, 10000 examples) from the training dataset and split the remaining data in a validation dataset of 500 examples and an unlabeled dataset U. We compare the methods on the soft-distillation. For each trial we train the student model for 40 epochs and keep the best test accuracy over all epochs. We perform 3 trials and report the average of each method and the variance of the achieved accuracies over the trials. The results of our experiments can be found in Table 8. We remark that we did not implement the UPS method for this dataset as the data-augmentation method for estimating the teacher's accuracy could not be readily used for this NLP dataset. Moreover, using dropout and Monte Carlo estimation for the uncertainty was also not compatible with the Albert model used in this experiment.

Since we are dealing with ALBERT-models (which are already pre-trained), we do not pre-train the student model on dataset A except in the case of "weighted-distillation" [27], where we pre-train the student model on dataset A just for 1 epoch. The teacher model is trained using the Adam optimizer for 20 epochs with initial learning rate $2 \cdot 10^{-5}$. The student model is trained also using the Adam optimizer but for 40 epochs and with learning rate $10^{-7}$.

The hyperparameters of each method are optimized as follows. For SLaM we do a hyperparameter search over $\{0.5, 0.6, 0.7, 0.8, 0.9\}$ for the lower bound for isotonic regression, and we keep the best performing value for each potential size of dataset $A$. As this is a binary classification benchmark we naturally set $k(x) = 2$ for all examples. For weighted distillation we do a grid search over updating the weights every $\{1, 10, 20, 40\}$ epochs and, similarly, we report the best average accuracy achieved. Finally, for VID (recall also Remark D.1) we search over $\{0.1, 0.5, 1.0, 2.0\}$ for the coefficient of

Table 8: Experiments on the Large Movies Reviews Dataset (**soft**-distillation). See Section D.3 for details.

| Labeled Examples | 2% | 4% | 8% | 40% |
|---|---|---|---|---|
| Teacher | 77.52 | 84.04 | 85.44 | 88.3 |
| Vanilla | $80.93 \pm 0.10$ | $85.12 \pm 0.29$ | $85.99 \pm 0.08$ | $87.50 \pm 0.6$ |
| Taylor-CE [20] | $79.5 \pm 0.38$ | $85.14 \pm 0.13$ | $85.98 \pm 0.14$ | $87.57 \pm 0.3$ |
| VID [3] | $81.76 \pm 0.32$ | $85.33 \pm 0.35$ | $86.17 \pm 0.06$ | $87.71 \pm 0.01$ |
| Weighted [27] | $81.1 \pm +0.1$ | $85.2 \pm 0.05$ | $86.13 \pm 0.17$ | $\mathbf{87.8 \pm 0.25}$ |
| SLaM (Ours) | $\mathbf{81.88 \pm 0.23}$ | $\mathbf{85.5 \pm 0.09}$ | $\mathbf{86.23 \pm 0.13}$ | $87.73 \pm 0.38$ |

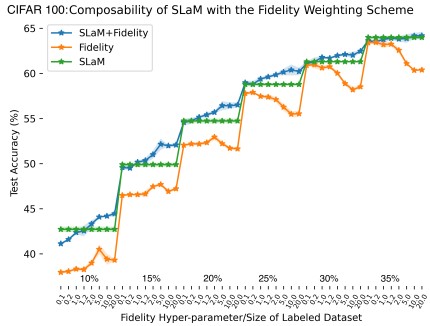

Figure 6: Composability the fidelity-based weighting scheme of [17]. The $x$-axis shows the different values of the fidelity hyper-parameter $\beta$ and the size of dataset A. From left to right we increase the size of dataset A from $10\%$ to $35\%$ and for each size we try different values of $\beta$. We observe that SLaM on its own (shown in green) is usually much better than the fidelity weighting scheme (shown in orange). Moreover, using SLaM on top of the fidelity weighting scheme (shown in blue) consistently improves its performance.

the VID-related term of the loss function, and for the PolyLoss we opitmize its hyperparameter over $\{-1.0, -0.8, -0.6, -0.4, -0.2, 0.5, 1.0, 2.0\}$.

### D.4 Combining with Teacher-Uncertainty-Based Reweighting Techniques

As we discussed in Section 2, our method can in principle be combined with teacher-uncertainty filtering and weighting schemes as these can be seen as preprocessing steps. To demonstrate this, we combine our method with the so-called fidelity-based weighting scheme of [17]. The fidelity weighting scheme reweights examples using some uncertainty measure for teacher's labels, e.g., by performing random data-augmentations and estimating the variance of the resulting teacher labels or using dropout and Monte Carlo estimation. More precisely, for every example $x$ in the teacher-labeled dataset $B$, the fidelity-weighting scheme assigns the weight $w^{\text{Fid}}(x) = \exp(-\beta \, \text{uncertainty}^{\text{teacher}}(x))$ for some hyper-parameter $\beta > 0$. In our experiments we performed $10$ random data augmentations (random crop and resize), estimated the coordinate-wise variance of the resulting teacher soft-labels, and finally computed the average of the variances of the $k$-classes, as proposed in [17]. We normalized the above uncertainty of each example by the total uncertainty of the teacher over the whole dataset $B$. The weights of examples in dataset $A$ are set to $1$ and the reweighted objective is optimized over the combination of the datasets $A, B$.

$$\mathcal{L}^{\text{fid}}(w) = \frac{1}{|A \cup B|} \left( \sum_{(x,y) \in A} \ell(y, f(x; w)) + \sum_{(x,y) \in B} w^{\text{Fid}}(x) \, \ell(y, f(x; w)) \right). \qquad (3)$$

To demonstrate the composability of our method with such uncertainty-based weighting schemes, we use CIFAR100 and the percentage of the labeled dataset A (as a fraction of the whole training set) is $10\%, 15\%, 20\%, 25\%, 30\%, 35\%$, similar to the setting of Section 4.2. The teacher is a ResNet110 and the student is a ResNet56. We first train the student using only the fidelity weighting scheme, i.e., optimize the loss function of Equation (4) using different values for the hyperparameter $\beta \in \{0.1, 0.2, 1.0, 1.2, 2.0, 5.0, 10.0, 20.0\}$, i.e., ranging from mildly reweighting the examples of

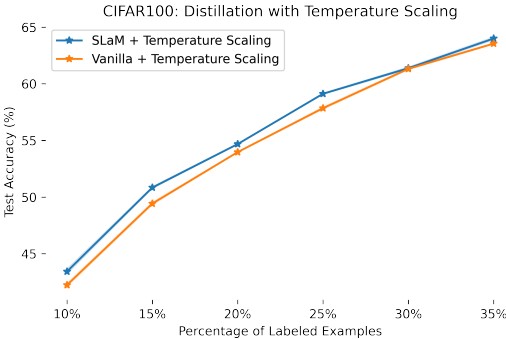

Figure 7: CIFAR100: Temperature Ablation. On the x-axis we have the size of the labeled dataset (as a percentage of the whole training dataset) that the teacher model uses for training.

dataset B to more agressively "removing" examples where the teacher's entropy is large. For the same values of $\beta$ we then train the student using the reweighted SLaM objective:

$$\mathcal{L}^{\text{Fid+SLaM}}(w) = \frac{1}{|A \cup B|} \left( \sum_{(x,y) \in A} \ell(y, f(x; w)) + \sum_{(x,y) \in B} w^{\text{fid}}(x) \, \ell(y, \text{mix}(f(x; w); \alpha(x), k(x))) \right).$$

(4)

For the combined SLaM + Fidelity method we did not perform hyper-parameter search and used the same parameters for the isotonic regression as we did in the "standard" SLaM experiment in CIFAR100 of Appendix D.1. We present our comprehensive results for all sizes of dataset A and values of the hyper-parameter $\beta$ in Figure 6. Our results show that, regardless of the value of the hyperparameter $\beta$ and the size of the labeled dataset A, using SLaM together with the fidelity weighting scheme provides consistent improvements. Moreover, in Figure 6, we observe that by using SLaM the achieved accuracy depends less on the hyper-parameter $\beta$: since SLaM takes into account the fact that some of the teacher's predictions are incorrect, it is not crucial to down-weight them or filter them out.

### D.5 Using Distillation Temperature

In this section we show that our approach can be effectively combined with temperature-scaling [26]. Choosing the right distillation temperature often provides significant improvements. In our setting, the teacher provides much more confident predictions (e.g., soft-labels with high-margin) on dataset A (where the teacher was trained) compared to the teacher soft-labels of dataset B where the teacher is, on average, less confident. Given this observation, it is reasonable to use different distillation temperatures for dataset A and dataset B. We try different temperatures for dataset A and dataset B and perform vanilla distillation with temperature and also consider applying the temperature scaling before applying SLaM. For each size of dataset A we try pairs of temperatures $t_A, t_B \in \{0.01, 0.1, 0.5, 0.8, 1., 2., 5., 10., 100.\}$ and report the best accuracy achieved by vanilla distillation and the best achieved by first applying temperature scaling and then SLaM. In Figure 7 we observe that SLaM with temperature scaling consistently improves over vanilla distillation with temperature.

### D.6 Using SLaM with other loss functions beyond cross-entropy

In this section, we demonstrate that our method can be successfully applied when the student loss function comes from the families of losses introduced in [20] and [35]. We perform experiments on CIFAR-100 and ImageNet following the setting of Section 4.2. In particular, we compare vanilla distillation with unlabeled examples using the Taylor-CE loss of [20] and the PolyLoss of [35], with combining SLaM with these losses. For the Taylor-CE loss we set the "degree" hyperparameter to be 2 (as suggested in [20]) and we set the hyperparameter of the PolyLoss to be 2.0 (as suggested in [35]). The corresponding results can be found in Figure 8.

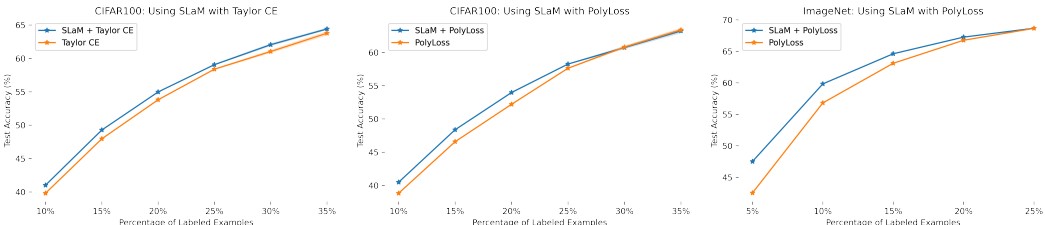

Figure 8: Using SLaM with PolyLoss [35] and Taylor CE [20]. On the x-axis we have the size of the labeled dataset (as a percentage of the whole training dataset) that the teacher model uses for training. See Appendix D.6 for more details.

## D.7 Performance of SLaM with (even) fewer labels

In this section we investigate more extensively the effect of the range of the the size of the labeled-dataset. In particular, we provide experiments with even fewer labeled examples available. Experiments in which the number of available labeled examples is small are of greater importance since this is the typical scenario where "distillation with unlabeled examples" applies. For larger dataset sizes, as seen in the plots of the previous experiments, all methods converge to roughly the same performance.

| Labeled Examples | CIFAR10, 1% | CIFAR10, 5% | CIFAR100, 1% | CIFAR100, 5% |
|---|---|---|---|---|
| Teacher | 10.07 | 51.67 | 9.45 | 23.61 |
| Vanilla | $11.75 \pm 0.1$ | $54.2 \pm 0.16$ | $10.08 \pm 0.06$ | $25.15 \pm 0.11$ |
| Taylor-CE [20] | $10.00 \pm 0.01$ | $55.14 \pm 0.28$ | $9.79 \pm 0.13$ | $26.14 \pm 0.39$ |
| UPS [48] | $12.74 \pm 0.94$ | $56.21 \pm 0.23$ | $10.40 \pm 0.05$ | $26.41 \pm 0.13$ |
| VID [3] | $13.25 \pm 0.26$ | $54.32 \pm 0.05$ | $10.02 \pm 0.13$ | $24.93 \pm 0.19$ |
| Weighted [27] | $12.67 \pm 0.01$ | $54.58 \pm 0.1$ | $10.07 \pm 0.06$ | $25.36 \pm 0.14$ |
| SLaM | $\mathbf{26.73 \pm 0.02}$ | $\mathbf{57.40 \pm 0.05}$ | $\mathbf{10.87 \pm 0.07}$ | $\mathbf{27.76 \pm 0.19}$ |

Table 9: Comparison of methods on CIFAR10 and CIFAR100 datasets.

## D.8 SLaM Hyper-parameter Ablation

We investigate the effect of the hyper-parameters used in SLaM, see Appendix D.1.

In Table 10 we investigate using different values for the isotonic regression lower-bound parameter, (lb in Appendix D.1). We observe that SLaM is rather robust to this hyperpameter and usually simply setting $lb = 0.5$ yields good results.

| Labeled Examples | 10% | 15% | 20% | 25% | 30% | 35% |
|---|---|---|---|---|---|---|
| lb=0 | $65.89 \pm 0.14$ | $71.32 \pm 0.11$ | $74.02 \pm 0.19$ | $76.71 \pm 0.08$ | $77.76 \pm 0.12$ | $78.9 \pm 0.12$ |
| lb=0.1 | $65.7 \pm 0.06$ | $71.73 \pm 0.17$ | $74.18 \pm 0.14$ | $76.42 \pm 0.10$ | $78.07 \pm 0.19$ | $\mathbf{78.99 \pm 0.15}$ |
| lb=0.3 | $65.88 \pm 0.21$ | $\mathbf{71.94 \pm 0.08}$ | $74 \pm 0.09$ | $\mathbf{76.76 \pm 0.13}$ | $\mathbf{78.19 \pm 0.18}$ | $79.01 \pm 0.21$ |
| lb=0.5 | $\mathbf{66.13 \pm 0.18}$ | $\mathbf{71.96 \pm 0.21}$ | $\mathbf{74.27 \pm 0.12}$ | $\mathbf{76.73 \pm 0.07}$ | $78.01 \pm 0.14$ | $\mathbf{78.98 \pm 0.09}$ |
| lb=0.7 | $64.69 \pm 0.09$ | $71.19 \pm 0.13$ | $73.74 \pm 0.18$ | $76.38 \pm 0.11$ | $\mathbf{78.2 \pm 0.10}$ | $78.71 \pm 0.16$ |
| lb=0.9 | $63.51 \pm 0.17$ | $69.86 \pm 0.14$ | $72.57 \pm 0.20$ | $75.32 \pm 0.17$ | $77.35 \pm 0.09$ | $78.19 \pm 0.11$ |
| lb=0.99 | $62.86 \pm 0.21$ | $69.84 \pm 0.18$ | $72.34 \pm 0.16$ | $75.16 \pm 0.21$ | $77.37 \pm 0.20$ | $78.13 \pm 0.13$ |
| lb=1 | $62.78 \pm 0.11$ | $69.84 \pm 0.09$ | $72.63 \pm 0.14$ | $75.1 \pm 0.19$ | $77.19 \pm 0.08$ | $78.17 \pm 0.16$ |

Table 10: CIFAR10: $a(x)$ ablation results for the lower bound lb of isotonic regression. The value of lb ranges from 0 to 1.

We next compare using a fixed value for the top-k threshold value, $k(x)$ versus the data-depenent method described in Appendix D.1. We again observe that SLaM is robust to the value of $k$ used since it outperforms vanilla distillation for reasonable values of $k$. Overall, we found that using fixed values of $k(x)$ (after some hyper-parameter search for $k$) and using the data-dependent method yield comparable results. The advantage of the fixed-value method is that it is easier to implement (and slightly more efficient) and the advantage of the data-dependent method is that its hyperparameter (threshold $t$ in Algorithm 2) is easier to tune (in all our experiments $t = 0.9$ achieved good performance).

| Labeled Examples | 10% | 15% | 20% |
|---|---|---|---|
| Vanilla | $37.94 \pm 0.10$ | $46.42 \pm 0.24$ | $52.17 \pm 0.21$ |
| k=2 | $41.25 \pm 0.36$ | $49.18 \pm 0.19$ | $54.48 \pm 0.25$ |
| k=5 | $40.71 \pm 0.29$ | $49.41 \pm 0.23$ | $54.41 \pm 0.2$ |
| k=10 | $41.2 \pm 0.8$ | $49.31 \pm 0.12$ | $54.42 \pm 0.19$ |
| Data-Dependent k(x) $(t = 0.9)$ | $\mathbf{42.7 \pm 0.30}$ | $\mathbf{49.89 \pm 0.23}$ | $\mathbf{54.73 \pm 0.27}$ |

Table 11: CIFAR100: Fixed-Value vs Data-Dependent $k(x)$ Ablation.

## D.9 Robustness of SLaM to inaccuracies in $\alpha(x), k(x)$

We perform a "controlled" experiment by adding noise to the estimates of $\alpha(x)$ and $k(x)$ to test the robustness of SLaM to inaccurate predictions of $\alpha(x)$ and $k(x)$ is a valuable addition to our experimental evaluation. To do this, we start from the oracle values for $\alpha(x)$ and $k(x)$, i.e., $\alpha^*(x) = 1$ if the teacher prediction is correct on $x$ and 0 if it is incorrect and $k^*(x)$ is equal to the smallest integer value $\ell$ so that the ground-truth label is contained in the top $\ell$ predictions of the teacher. We then introduce random noise to the oracle predictions: for $\alpha(x)$ we perform a random perturbation

$$\alpha(x) = \alpha^*(x) + (1 - 2\alpha^*(x))\, \xi,$$

where $\xi$ is uniformly distributed in $[0, \sigma]$. Hence, when $\alpha^*(x) = 0$ we increase it by adding a random variable $\xi \in [0, \sigma]$ and when $\alpha^*(x) = 1$ we decrease it by subtracting the same random variable $\xi \in [0, \sigma]$. We clip the resulting value in the interval $[0, 1]$. To create noisy values for $k(x)$ we simply add a random integer to the optimal value $k^*(x)$, i.e., $k(x) = k^*(x) + Z$, where $Z$ is a random integer in $\{-\ell, \dots, \ell\}$. We clip the resulting value of $k(x)$ in the range $\{0, \dots, \text{numClasses}\}$.

We test our method on CIFAR100 and observe that the accuracy of SLaM gracefully decays as the predictions for $\alpha(x)$ and $k(x)$ become worse (bottom right corner is the noisiest $\sigma = 0.5, \ell = 90$ and top left is the noiseless $\sigma = 0, \ell = 0$), see Table 12.

| CIFAR100, 20% Labeled Data | $\ell = 0$ | $\ell = 5$ | $\ell = 10$ | $\ell = 50$ | $\ell = 90$ |
|---|---|---|---|---|---|
| $\sigma = 0$ | $61.69 \pm 0.3$ | $59.71 \pm 0.49$ | $59.7 \pm 0.55$ | $59.63 \pm 0.44$ | $59.4 \pm 0.65$ |
| $\sigma = 0.1$ | $60.04 \pm 0.29$ | $59.55 \pm 0.33$ | $59.66 \pm 0.35$ | $59.79 \pm 0.2$ | $60.03 \pm 0.41$ |
| $\sigma = 0.2$ | $59.39 \pm 0.1$ | $59.23 \pm 0.25$ | $59.5 \pm 0.39$ | $59.16 \pm 0.29$ | $59.31 \pm 0.31$ |
| $\sigma = 0.5$ | $57.71 \pm 0.15$ | $57.6 \pm 0.23$ | $57.56 \pm 0.28$ | $57.46 \pm 0.25$ | $57.29 \pm 0.21$ |

Table 12: CIFAR100 with 20% labeled data performance with different levels of noise added to the predictions for $k(x)$ and $\alpha(x)$.

## D.10 Different Regression Algorithms for $\alpha(x), k(x)$

In our main experimental evaluation of SLaM, we chose isotonic regression to enforce the monotonicity in the learned accuracy estimates for $\alpha(x)$ based on the empirical observation that $\alpha(x)$ is often approximately monotone as a function of the margin of the teacher. Moreover, the lower threshold (denoted by lb in Appendix D.1) in isotonic regression gives us a way to control how "agressive" the mixing operation is going to be. That said, SLaM does not hinge on some particular regression method and other methods can be used. We investigate using different regression methods (knn, Logistic regression) for estimating $\alpha(x), k(x)$ with SLaM. In Table 13 and Table 14, we see that Isotonic regression typically outperfoms other methods. Moreover, SLaM provides consistent improvements regardless of regression method used.

| Labeled Data | 10% | 15% | 20% | 25% | 30% | 35% |
|---|---|---|---|---|---|---|
| **kNN** $k=10$ | $67.1 \pm 0.15$ | $70.56 \pm 0.21$ | $74.6 \pm 0.11$ | $76.68 \pm 0.17$ | $78 \pm 0.16$ | $79.26 \pm 0.12$ |
| **kNN** $k=20$ | $67.5 \pm 0.15$ | $71.09 \pm 0.19$ | $74.47 \pm 0.15$ | $77.03 \pm 0.1$ | $78.03 \pm 0.17$ | $79.21 \pm 0.13$ |
| **kNN** $k=30$ | $67.51 \pm 0.11$ | $71.27 \pm 0.13$ | $74.66 \pm 0.12$ | $77.03 \pm 0.13$ | $78.07 \pm 0.2$ | $79.2 \pm 0.18$ |
| **kNN** $k=40$ | $67.64 \pm 0.21$ | $71.08 \pm 0.22$ | $74.5 \pm 0.09$ | $76.64 \pm 0.12$ | $77.92 \pm 0.11$ | $79.41 \pm 0.22$ |
| **Logistic** | $65.26 \pm 0.05$ | $68.85 \pm 0.08$ | $73.35 \pm 0.12$ | $76.17 \pm 0.15$ | $76.87 \pm 0.25$ | $78.76 \pm 0.35$ |
| **Isotonic** $lb=0.5$ | $66.82 \pm 0.61$ | $72.61 \pm 0.30$ | $75.01 \pm 0.25$ | $75.72 \pm 0.17$ | $78.04 \pm 0.16$ | $79.22 \pm 0.11$ |

Table 13: CIFAR10: Using different regression methods for estimating $\alpha(x), k(x)$.

| Labeled Data | 10% | 15% | 20% | 25% | 30% | 35% |
|---|---|---|---|---|---|---|
| **kNN** *k=10* | $40.75 \pm 0.02$ | $49.07 \pm 0.15$ | $54.86 \pm 0.11$ | $57.87 \pm 0.17$ | $61.9 \pm 0.2$ | $63.06 \pm 0.22$ |
| **kNN** *k=20* | $41.03 \pm 0.05$ | $49.19 \pm 0.12$ | $54.9 \pm 0.1$ | $57.85 \pm 0.18$ | $61.76 \pm 0.2$ | $63.45 \pm 0.16$ |
| **kNN** *k=30* | $41.04 \pm 0.07$ | $49.55 \pm 0.15$ | $55.14 \pm 0.15$ | $57.96 \pm 0.21$ | $61.9 \pm 0.19$ | $63.2 \pm 0.23$ |
| **kNN** *k=40* | $41.23 \pm 0.03$ | $49.76 \pm 0.15$ | $54.78 \pm 0.1$ | $58.15 \pm 0.17$ | $61.98 \pm 0.21$ | $63.47 \pm 0.2$ |
| **Logistic** | $39.68 \pm 0.03$ | $48.17 \pm 0.1$ | $53.56 \pm 0.11$ | $57.45 \pm 0.09$ | $61.77 \pm 0.19$ | $63.24 \pm 0.18$ |
| **Isotonic** *lb=0.5* | $42.72 \pm 0.30$ | $49.89 \pm 0.23$ | $54.73 \pm 0.27$ | $58.78 \pm 0.15$ | $61.30 \pm 0.09$ | $63.98 \pm 0.19$ |

Table 14: CIFAR100: Using different regression methods for estimating $\alpha(x), k(x)$.

## D.11 Validation dataset size ablation

In this section we investigate the effect of the size of the validation dataset required by SLaM. As we have already showed in our previous experiments, SLaM requires only rough estimates of $\alpha(x)$ and $k(x)$ and thus even very small validation datasets suffice. We observe that SLaM is able to provide improvements even with very small validation datasets (e.g., with 128 labels).

In Table 15 we use different validation sizes for the CIFAR-100 experiment described in our manuscript and and show that the performance of SLaM improves when the validation dataset is larger but the gaps are not very significant especially for larger sizes of the labeled dataset. We show that SLaM is able to provide improvements even with very small validation datasets (e.g., with 128 labels).

| Labeled Data | 10% | 15% | 20% | 25% | 30% | 35% |
|---|---|---|---|---|---|---|
| 128 | $40.67 \pm 0.24$ | $48.95 \pm 0.18$ | $54.27 \pm 0.21$ | $58.87 \pm 0.19$ | $61.42 \pm 0.22$ | $63.65 \pm 0.21$ |
| 256 | $40.97 \pm 0.12$ | $49.21 \pm 0.17$ | $54.18 \pm 0.23$ | $58.54 \pm 0.13$ | $61.18 \pm 0.25$ | $63.19 \pm 0.07$ |
| 512 | $41.06 \pm 0.30$ | $49.27 \pm 0.19$ | $54.36 \pm 0.12$ | $58.57 \pm 0.26$ | $61.25 \pm 0.31$ | $63.38 \pm 0.11$ |
| 1024 | $41.83 \pm 0.32$ | $49.35 \pm 0.25$ | $54.71 \pm 0.18$ | $58.95 \pm 0.32$ | $61.28 \pm 0.46$ | $63.62 \pm 0.28$ |

Table 15: CIFAR-100 validation dataset size ablation results.

## E    Distilling Linear Models and Learning Noisy Halfspaces

In this section we state and prove our convergence result for the SLaM method when applied to linear models. Our assumption is that the ground-truth $g(x)$ corresponds to a halfspace, i.e., $g(x) = (\mathbf{1}\{w^* \cdot x > 0\}, \mathbf{1}\{w^* \cdot x \leq 0\})$ for some unknown weight vector $w^*$. We show that using SLaM with a linear model as the student will recover the ground truth classifier. We make the standard assumption that the ground-truth halfspace has $\gamma$-margin, i.e., that $\|w^*\|_2 = 1$ and that it holds $|w^* \cdot x| \geq \gamma$ for all examples $x$. For a fixed example $x$, the observed noisy teacher-label $y$ satisfies Definition 3.2, i.e., $y = g(x)$ w.p. $\alpha(x)$ and $y = 1 - g(x)$ w.p. $1 - \alpha(x)$ (since $k = 2$ for binary classification). Our approach consists of using the standard cross-entropy loss $\mathrm{ce}(p, q)$ and training a student-model consisting of a linear layer plus a soft-max activation, i.e.,

$$ f(x; w) = (f_0(x; w), f_1(x; w)) = \left( \frac{1}{1 + e^{-w \cdot x}}, \frac{e^{-w \cdot x}}{1 + e^{-w \cdot x}} \right) . $$

Recall, that for binary classification, we define the mixing operation as

$$ \mathrm{mix}(f(x; w); \alpha(x)) = \alpha(x) f(x; w) + (1 - \alpha(x))(1 - f(x; w)) . $$

**Theorem E.1** (Student Label Mixing Convergence). *Let $X$ be a distribution on $\mathbf{R}^d$ and $g(x)$ be the ground-truth halfspace with normal vector $w^* \in \mathbf{R}^d$. Let $D$ be the distribution over (noisy) teacher-labeled examples $(x, y)$ whose $x$-marginal is $X$. We denote by $\alpha(x)$ the probability that the teacher label $y \in [0, 1]^2$ is correct, i.e., $\alpha(x) = \mathbf{P}_{(x,y) \sim D}[\mathrm{argmax}(y) = g(x) \mid x]$. Assume that there exist $\beta, \gamma > 0$ such that for all examples $x$ in the support of $X$ it holds that $|w^* \cdot x| \geq \gamma$ and $|1/2 - \alpha(x)| \leq \beta$. Let $\epsilon > 0$. After $T = O(1/(\beta^2 \gamma^2 \epsilon^2))$ iterations of SLaM (Algorithm 3), with probability at least $99\%$, there exists an iteration $t \leq T$ where $\mathbf{P}_{x \sim X}[\mathrm{err}(f(x; w^{(t)}), g(x))] \leq \epsilon$.*

*Remark* E.2 (High-Probability Result). We remark that even though our learner succeeds with constant probability (at least %99) we can amplify its success probability to $1 - \delta$ by standard amplification techniques (i.e., by repeating the algorithm $O(\log(1/\delta))$ times and keeping the best result). To achieve success probability $1 - \delta$ the total sample complexity is $O(\log(1/\delta)/(\epsilon^2 \gamma^2 \beta^2))$.

*Proof.* We first provide simplified expressions for the gradient of the SLaM objective and the update vectors $\lambda^{(t)} g^{(t)}$ used in Algorithm 3. In what follows we remark that for any binary classification

---

**Algorithm 3** SLaM for Linear Models

---

Initialiaze weight vector of student $w^{(0)} \leftarrow 0$
**for** $t = 1, \ldots, T$ **do**
    Draw example $x^{(t)} \sim X$.
    Label $x^{(t)}$ with (noisy) teacher to obtain $y^{(t)}$
    Compute the gradient of the SLaM loss at $(x^{(t)}, y^{(t)})$:

$$g^{(t)} \leftarrow \partial_w \mathrm{ce}(y^{(t)}, \mathrm{mix}(f(x^{(t)}); w^{(t-1)}), \alpha(x^{(t)})) \mid_{w=w^{(t-1)}}$$

    Compute step size: $\lambda^{(t)} \leftarrow 1/r(f(x^{(t)}; w^{(t-1)}), \alpha(x^{(t)}))$ (see Lemma E.3 for the definition of $r(\cdot, \cdot)$).
    Update the student model: $w^{(t)} \leftarrow w^{(t-1)} - \lambda^{(t)} \, g^{(t)}$
**end for**

---

model $f(x; w) = (f_0(x; w), f_1(x; w))$ we have the following identities: (i) $(\mathrm{mix}(f(x; w); \alpha(x)))_0 = \mathrm{mix}(f_0(x; w); \alpha(x))$, where to simplify notation we overload the mixing operation to also act on the scalar $f_0(x; w)$, i.e., $\mathrm{mix}(f_0(x; w); \alpha(x)) = \alpha(x) f_0(x; w) + (1 - \alpha(x))(1 - f_0(x; w))$; and (ii) $f_1(x; w) = 1 - f_0(x; w)$.

**Lemma E.3** (SLaM Gradient). *The gradient of the SLaM objective is equal to*

$$\partial_w \mathrm{ce}(y, \mathrm{mix}(f(x; w); \alpha(x)) = r(f_0(x; w); \alpha(x)) \, \mathrm{sgn}(2\alpha(x) - 1) \, ((\mathrm{mix}(f_0(x; w); \alpha(x)) - y_0)x,$$

*where*

$$r(f(x; w); \alpha(x)) = \frac{f_0(x; w)(1 - f_0(x; w))}{\mathrm{mix}(f_0(x; w); \alpha(x))(1 - \mathrm{mix}(f_0(x; w), \alpha(x)))} \, |2\alpha(x) - 1|$$

*Let $L(x; w) = \mathbf{E}_{(x,y)\sim D}[\mathrm{ce}(y, \mathrm{mix}(f(x; w), \alpha(x)) \mid x]$ be the expected student label mixing loss conditional on some example $x \in \mathbf{R}^d$. It holds $\partial_w L(x; w) = r(f(x; w), \alpha(x)) \, |2\alpha(x) - 1| \, (f_0(x; w) - g_0(x)) \, x$.*

*Proof.* We first show the formula

$$\partial_w \mathrm{ce}(y, \mathrm{mix}(f(x; w), \alpha(x)) = r(f_0(x; w), \alpha(x)) \, \mathrm{sgn}(2\alpha(x) - 1) \, ((\mathrm{mix}(f_0(x; w), \alpha(x)) - y_0)x \,. \tag{5}$$

Using the chain rule, we obtain

$$\partial_w \mathrm{ce}(y, \mathrm{mix}(f(x; w); \alpha(x)) =$$
$$- \frac{y_0}{\mathrm{mix}(f_0(x; w), \alpha(x))} \partial_w(\mathrm{mix}(f_0(x; w); \alpha(x))$$
$$- \frac{y_1}{\mathrm{mix}(f_1(x; w), \alpha(x))} \partial_w(\mathrm{mix}(f_1(x; w); \alpha(x)) \,.$$

Now we observe that that for binary classification, it holds that $y_1 = 1 - y_0$, $\mathrm{mix}(f_1(x; w); \alpha(x)) = 1 - \mathrm{mix}(f_0(x; w); \alpha(x))$, and therefore, also $\partial_w \mathrm{mix}(f(x; w); \alpha(x))_1) = -\partial_w \mathrm{mix}(f(x; w); \alpha(x))_0)$ to obtain the simplified expression:

$$\partial_w \mathrm{ce}(y, \mathrm{mix}(f(x; w); \alpha(x)) =$$
$$- \frac{y_0}{\mathrm{mix}(f_0(x; w), \alpha(x))} \partial_w(\mathrm{mix}(f_0(x; w); \alpha(x))$$
$$+ \frac{1 - y_0}{1 - \mathrm{mix}(f_0(x; w), \alpha(x))} \partial_w(\mathrm{mix}(f_0(x; w); \alpha(x)) \,.$$

Further simplifying the above expression, we obtain:

$$\partial_w \mathrm{ce}(y, \mathrm{mix}(f(x; w); \alpha(x)) =$$
$$= \frac{\mathrm{mix}(f_0(x; w), \alpha(x)) - y_0}{\mathrm{mix}(f_0(x; w), \alpha(x)) \, (1 - \mathrm{mix}(f_0(x; w), \alpha(x)))} \partial_w(\mathrm{mix}(f_0(x; w); \alpha(x)) \,.$$

Using again the chain rule we obtain that

$$\partial_w(\text{mix}(f_0(x;w);\alpha(x)) = \alpha(x)\partial_w(f_0(x;w))+(1-\alpha(x))\partial_w(1-f_0(x;w)) = (2\alpha(x)-1)\,\partial_w f_0(x;w)\,.$$

Using the fact that the derivative of the sigmoid function $r(t) = 1/(1+e^{-t})$, is $r'(t) = e^{-t}/(1-e^{-t})^2 = r(t)(1-r(t))$, and the chain rule, we obtain that $\partial_w f_0(x;w) = f_0(x;w)(1-f_0(x;w))x$. Putting everything together we obtain the claimed formula for $\partial_w\text{ce}(y,\text{mix}(f(x;w);\alpha(x)))$.

To obtain the gradient formula for the expected loss conditional on some fixed example $x$, we can use the fact that $\partial_w\,\mathbf{E}[\text{ce}(y,\text{mix}(f(x;w);\alpha(x)))\mid x] = \mathbf{E}[\partial_w\text{ce}(y,\text{mix}(f(x;w);\alpha(x)))\mid x]$. Now using the formula of Equation (5) and the fact that $\mathbf{E}[y_0\mid x] = \text{mix}(g_0(x);\alpha(x))$ by the definition of our noise model, we obtain that

$$\partial_w L(x;w) = r(f_0(x;w);\alpha(x))\text{sgn}(2\alpha(x)-1)(\text{mix}(f_0(x;w);\alpha(x)) - \text{mix}(g_0(x);\alpha(x)))$$
$$= r(f_0(x;w);\alpha(x))(2\alpha(x)-1)(f_0(x;w)-g_0(x))$$

$$\square$$

We first show the following claim proving that after roughly $T = 1/(\beta^2\gamma^2\epsilon^2)$ gradient iterations the student parameter vector $w^{(t)}$ will have good correlation with the ground-truth vector $w^*$.

*Claim* 1. Fix any $T$ larger than a sufficiently large constant multiple of $\log(1/\delta)/(\epsilon^2\gamma^2\beta^2)$, and assume that for all $t \leq T$ it holds that $\mathbf{P}_{x\sim X}[\text{err}(f(x;w^{(t)}),g(x))] > \epsilon$. Then, we have $w^{(T)}\cdot w^* = \Omega(\beta\gamma\epsilon)\,T$, with probability at least $1-\delta$.

*Proof.* Denote by $u^{(t)} = -\lambda^{(t)}g^{(t)}$ the update vector used in Algorithm 3. We observe that the weight vector at round $T$ is equal to $w^{(T)} = \sum_{t=1}^{T} u^{(t)}$. In what follows we denote by $\mathcal{F}^{(t)}$ the filtration corresponding to the randomness of the updates of Algorithm 3. We define the martingale $q^{(T)} = \sum_{t=1}^{T}(\mathbf{E}[u^{(t)}\mid\mathcal{F}^{(t-1)}]-u^{(t)})$ with $q^{(0)} = 0$. We first show that under the assumption that $\mathbf{P}_{x\sim X}[\text{argmax}(f(x;w^{(t)})) \neq g(x)] > \epsilon$, for all $t \leq T$, it holds that $\sum_{t=1}^{T}\mathbf{E}[u^{(t)}\mid\mathcal{F}^{(t-1)}]\cdot w^* \geq (\epsilon\gamma\beta/2)\,T$. Using the SLaM gradient expression of Lemma E.3 and the definition of the step size $\lambda^{(t)}$ we obtain that $\mathbf{E}[u^{(t)}\mid\mathcal{F}^{(t-1)}] = \mathbf{E}_{x\sim X}[|2\alpha(x)-1|\,(g_0(x)-f_0(x;w^{(t-1)}))\,x]$. Take any step $t$. We have that

$$\mathbf{E}[u^{(t)}\mid\mathcal{F}^{(t-1)}]\cdot w^* = \mathbf{E}_{x\sim X}[|2\alpha(x)-1|\,(g_0(x)-f_0(x;w^{(t-1)}))\,(x\cdot w^*)]$$
$$= \mathbf{E}_{x\sim X}[|2\alpha(x)-1|\,|g_0(x)-f_0(x;w^{(t-1)})|\,|x\cdot w^*|]\,,$$

where we used the fact that $(g_0(x)-f_0(x;w^{(t-1)}))\,\text{sgn}(x\cdot w^*) = |g_0(x)-f_0(x;w^{(t-1)})|$. Now, using the $\gamma$-margin assumption of the distribution $D$ and the fact that $|2\alpha(x)-1| \geq \beta$ we obtain

$$\mathbf{E}[u^{(t)}\mid\mathcal{F}^{(t-1)}]\cdot w^* \geq \beta\gamma\,\mathbf{E}_{x\sim X}[|g_0(x)-f_0(x;w^{(t-1)})|]$$
$$\geq \beta\gamma\,\mathbf{E}_{x\sim X}[|g_0(x)-f_0(x;w^{(t-1)})|\,\text{err}(g(x),f(x;w^{(t-1)}))]$$
$$\geq (\beta\gamma/2)\,\mathbf{P}_{x\sim X}[\text{err}(g(x),f(x;w^{(t-1)}))] \geq \beta\gamma\epsilon/2\,,$$

where for the penultimate inequality we used the fact that when $g(x)$ and $f(x;w^{(t-1)})$ disagree it holds that $|g_0(x)-f_0(x;w^{(t-1)})| \geq 1/2$. Take, for example, the case where $g_0(x) = 1$. Then $f_0(x;w^{(t-1)})$ must be smaller than $1/2$ otherwise the prediction of the model $\text{argmax}\,f(x;w^{(t-1)})$ would also be 0 (and would agree with the prediction of $g(x)$). Finally, for the last inequality we used the fact that, by our assumption, it holds that $\mathbf{P}_{x\sim X}[\text{err}(g(x),f(x;w^{(t-1)}))] \geq \epsilon$. Therefore, we conclude that $\sum_{t=1}^{T}\mathbf{E}[u^{(t)}\mid\mathcal{F}^{(t-1)}]\cdot w^* \geq (\epsilon\gamma\beta/2)\,T$. Next, we shall show that $w^{(T)}$ also achieves good correlation with the optimal direction $w^*$ with high probability. We will use the fact that $q^{(t)}$ is a martingale and the Azuma-Hoeffding inequality to show that $w^{(T)}\cdot w^*$ will not be very far from its expectation.

**Lemma E.4** (Azuma-Hoeffding). *Let $\xi^{(t)}$ be a martingale with bounded increments, i.e., $|\xi^{(t)}-\xi^{(t-1)}| \leq M$. It holds that $\mathbf{P}[\xi^{(T)} \geq \xi^{(0)}+\lambda] \leq e^{-\lambda^2/(2M^2 T)}$.*

Recall that from Lemma E.3 we have that $\mathbf{E}[u^{(t)} \mid \mathcal{F}^{(t-1)}] = \mathbf{E}_{x \sim X}[|2\alpha(x) - 1| \, (g_0(x) - f_0(x; w^{(t-1)})) \, x]$ and

$$u^{(t)} = \text{sgn}(2\alpha(x^{(t)}) - 1) \, (y_0^{(t)} - \text{mix}(f_0(x^{(t)}; w^{(t-1)}), \alpha(x^{(t)}))) \, x^{(t)} \,.$$

Observe that since $\|x\|_2 \leq 1$ for all $x$ it holds that $\|u^{(t)}\|_2 \leq 1$. Therefore, the difference $\|\mathbf{E}[u^{(t)} \mid \mathcal{F}^{(t-1)}] - u^{(t)}\| \leq 2$ with probability 1. Since $\|w^*\|_2 = 1$, using Cauchy-Schwarz, we also obtain that $|\mathbf{E}[u^{(t)} \cdot w^* \mid \mathcal{F}^{(t-1)}] - u^{(t)} \cdot w^*| \leq 2$.

Using Lemma E.4, and the fact that $q^{(0)} = 0$ we obtain that $\mathbf{P}[q^{(t)} \cdot w^* \geq (\beta\gamma\epsilon/4) \, T] \leq e^{-\beta^2\gamma^2\epsilon^2 T/128}$. Therefore we conclude that for any $T$ larger than $128 \log(1/\delta)/(\beta^2\gamma^2\epsilon^2)$, with probability at least $1 - \delta$, it holds that $q^{(T)} \cdot w^* \geq (\beta\gamma\epsilon/4)T$ or equivalently $w^{(T)} \cdot w^* \geq (\beta\gamma\epsilon/4) \, T$, where we used our previously obtained bound for the expected updates $\sum_{t=1}^{T} \mathbf{E}[u^{(t)} \mid \mathcal{F}^{(t-1)}] \cdot w^* \geq (\beta\gamma\epsilon/2) \, T$.

$\square$

*Claim 2.* Fix any $T \geq 1$. Then, we have $\|w^{(T)}\|_2 = O(\sqrt{T})$, with probability at least 99%.

*Proof.* We have that $\|w^{(T)}\|_2^2 = \|w^{(T-1)}\|_2^2 + 2u^{(T)} \cdot w^{(T-1)} + \|u^{(T)}\|_2^2$. Unrolling the iteration, we obtain that

$$\|w^{(T)}\|_2^2 = 2\sum_{t=1}^{T} u^{(t)} \cdot w^{(t-1)} + \sum_{t=1}^{T} \|u^{(t)}\|_2^2 \leq 2\sum_{t=1}^{T} u^{(t)} \cdot w^{(t-1)} + T \,, \tag{6}$$

where we used the fact that, since $\|x^{(t)}\|_2 \leq 1$, it holds that $\|u^{(t)}\|_2 \leq 1$ (see the proof of Claim 1). We first show that $\sum_{t=1}^{T} \mathbf{E}[u^{(t)} \mid \mathcal{F}^{(t-1)}] \cdot w^{(t-1)} = O(T)$. We have

$$\mathbf{E}[u^{(t)} \mid \mathcal{F}^{(t-1)}] \cdot w^{(t-1)} = \mathbf{E}_{x \sim X}[|2\alpha(x) - 1| \, (g_0(x) - f_0(x; w^{(t-1)})) \, (x \cdot w^{(t-1)})]$$
$$\leq \mathbf{E}_{x \sim X}[(g_0(x) - f_0(x; w^{(t-1)})) \, (x \cdot w^{(t-1)})] \,.$$

We will show that for $x$ it holds that

$$g_0(x) - f(x; w^{(t-1)})(x \cdot w^{(t-1)}) \leq \frac{1}{e} \,.$$

Fix some $x$ and let $s = w^{(t-1)} \cdot x$. Assume first that $g_0(x) = 1$. Then, we have

$$g_0(x) - f(x; w^{(t-1)})(x \cdot w^{(t-1)}) = \left(1 - \frac{1}{1 + e^{-s}}\right) s = s \frac{e^{-s}}{1 + e^{-s}} \leq \frac{1}{e} \,,$$

where we used the fact that $s \frac{e^{-s}}{1+e^{-s}} \leq 0$ for $s \leq 0$ and $s \frac{e^{-s}}{1+e^{-s}} \leq se^{-s} \leq 1/e$ for $s \geq 0$ (using the elementary inequality $ze^{-z} \leq 1/e$ for all $z \in \mathbf{R}$). When $g_0(x) = 0$ we similarly have that

$$g_0(x) - f(x; w^{(t-1)})(x \cdot w^{(t-1)}) = -\frac{s}{1 + e^{-s}} \leq \frac{1}{e} \,,$$

where we used the fact that when $s \geq 0$ it holds that $-\frac{s}{1+e^{-s}} \leq 0$ and when $s \leq 0$, $-\frac{s}{1+e^{-s}} \leq -s/e^{-s} = -se^s$. For the final inequality, we used again the inequality $ze^{-z} \leq 1/e$ for all $z \in \mathbf{R}$ (where we replaced $z$ with $-s$).

Therefore, we obtain that $\mathbf{E}[u^{(t)} \mid \mathcal{F}^{(t-1)}] \cdot w^{(t-1)} \leq 1/e$ and $\sum_{t=1}^{T} \mathbf{E}[u^{(t)} \mid \mathcal{F}^{(t-1)}] \cdot w^{(t-1)} \leq T/e$. Using the decomposition of Equation (6), linearity of expectation, and the tower rule for conditional expectations, we conclude that $\mathbf{E}[\|w^{(T)}\|_2^2] \leq (2/e + 1)T$. Using Markov's inequality we obtain that with probability at least 99% it holds that $\|w^{(T)}\|_2^2 = O(T)$ or equivalently $\|w^{(T)}\|_2 = O(\sqrt{T})$.

$\square$

We can now finish the proof of Theorem 5.1. Assume, in order to reach a contradiction, that for all $t \leq T$ it holds that $\mathbf{P}_{x \sim X}[\text{err}(f(x; w^{(t)}), g(x))] > \epsilon$. Now picking $T$ to be larger than a sufficiently large constant multiple of $1/(\epsilon^2\gamma^2\beta^2)$ and using Claim 1 and Claim 2 we obtain that,

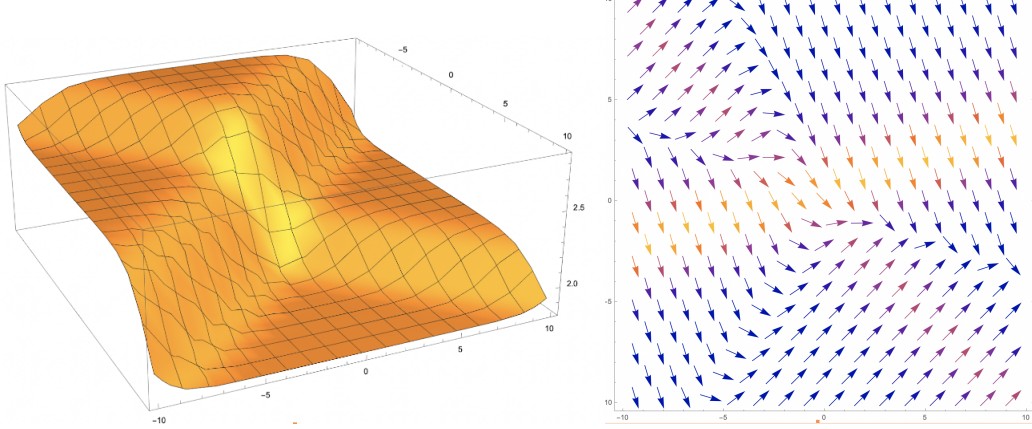

Figure 9: The landscape and gradient field of the population student label mixing loss for a simple 2 dimensional feature problem with a ground truth corresponding to a halfspace. We observe that the landscape is non-convex; however we can see that the corresponding gradient field "points towards the optimal direction" and therefore gradient descent converges to the global minimizer. A potential issue is the fact that the landscape contains regions where the gradients may almost vanish and this could lead to the gradient iteration of the student getting trapped there. To handle this issue, in Algorithm 3 we multiply the gradient of SLaM with an appropriate step-size.

with probability at least 99%, it holds that $w^{(T)} \cdot w^* / \|w^{(T)}\|_2 \geq \Omega(\beta\gamma\epsilon\sqrt{T})$, which can be made to be larger than 1 by our choice of $T$. However, this is a contradiction as by Cauchy-Schwarz we have $w^{(T)} \cdot w^* / \|w^{(T)}\|_2 \leq \|w^*\|_2 \leq 1$. Therefore, with probability at least 99%, it must be that for some $t \leq T$ it holds that $\mathbf{P}_{x \sim X}[\mathrm{err}(f(x; w^{(t)}), g(x))] \leq \epsilon$.

$\square$

