# OpenReview forum: "SLaM: Student-Label Mixing for  Distillation with Unlabeled Examples"
_NeurIPS.cc/2023/Conference — NeurIPS 2023 poster_

### Official Review · Reviewer_7BQA · 2023-06-21

**Soundness:** 3 good
**Presentation:** 3 good
**Contribution:** 3 good
**Rating:** 5
**Confidence:** 4

**Summary:**

This paper aims to systematically utilize the noise of teacher predictions in a knowledge distillation setting comprised of both labeled and unlabeled samples. The proposed method, SLaM linearly mixes the student predictions with the noise of the teacher predictions, thereby hoping that the student will learn the noise-free true underlying label. The method is supported by both theoretical results and empirical experiments on different datasets.

**Strengths:**

- The paper is well-written, easy to follow, and nicely structured.
- The proposed method, SLaM is relatively simple and general which is a nice feature.
- The method is supported by both theoretical and empirical results.

**Weaknesses:**

I am generally positive about the paper, but my biggest concern is the empirical evaluation of the methods. I am open to changing my score if the following is addressed.

Comments and questions:
- All empirical results presented are based on "[...] the best test accuracy over all epochs.". I.e. the methods are continuously evaluated on the test set (effectively making it a validation set), and no real test set performance is available - this is bad practice. Thus, I am concerned whether the reported scores are actually representative of the real test performance of the methods, and thereby if one can, in fact, rely on the conclusions drawn. I would strongly suggest reporting something like the test performance at the last epoch to avoid overfitting the test set. And even so, the experiments might be biased towards having evaluated the methods on the test set previously, so ideally evaluations on completely independent test sets would be included.
- In e.g. L230 SLaM is reported to "[...] consistently outperform the baselines, often by a large margin". I believe this is too bold a claim. In particular, only 7 of 24 experiments yield improvements over $1\%$ compared to the second-best baseline, while 8 other of 24 have improvements $<0.2\%$ of which 3 have improvements $<0.02\%$. Furthermore, there are 2 of 24 experiments where the method does not improve over the baselines. All of these observations are without consideration of the variance, which likely makes some additional improvements statistically indistinguishable. I would suggest reducing the claims slightly, especially given the evaluation protocol.
- It is unclear why this method shouldn't be considered a semi-supervised learning method, as both labeled and unlabeled samples are provided and used for the distillation procedure. In fact, the availability of a labeled validations set, $V$ appears to be a necessity for this method, and I believe it should be presented as a semi-supervised setting more explicitly.
- A lot of results and details are deferred to the appendix. While I fully understand the limitations a 9-page restriction can put on a paper, I believe some of the space in the paper could be better utilized to present at least some key results from the appendix.

Minor comments:
- L148: "agree" -> "disagree"
- L166: "etting" -> "setting"
- L311: "inFigure 2" -> "in Figure 2"
- Figure 1 is never referenced in the paper
- Figure 2 is badly formatted; i.e. too small a font and hard to distinguish the results/lines of different methods (consider rescaling the y-axis to e.g. improvement over baseline)

**Questions:**

See the above.

**Limitations:**

Sufficiently addressed.

---

> ### Author Rebuttal · Authors · 2023-08-10
>
>
>
> >All empirical results presented are based on "[...] the best test accuracy over all epochs... I would strongly suggest reporting something like the test performance at the last epoch to avoid overfitting the test set. ... completely independent test sets would be included.
>
> We agree with the reviewer that reporting the last-epoch accuracy is a better metric and are willing to change our tables to report that. We used the best test accuracy over all epochs to be consistent with the way the baselines of the prior works were evaluated.  In the following table we report the final epoch accuracy of all methods for CIFAR100. We observe that SLaM again consistently outperforms the baselines.
>
> Comparison of Last-Epoch Accuracy
> | CIFAR100 |  10%     |  15%     |  20%    |  25%    |  30%    |  35%    |
> |-|-|-|-|-|-|-|
> | Vanilla                           | $37.97 \pm 0.1$  | $46.37 \pm 0.12$ | $51.9 \pm 0.13$  | $57.85 \pm 0.17$ | $60.86 \pm 0.18$ | $63.56 \pm 0.35$ |
> | Taylor-CE                         | $40.12 \pm 0.03$ | $47.99 \pm 0.21$ | $54.18 \pm 0.21$ | $57.76 \pm 0.06$ | $61.22 \pm 0.11$ | $63.50 \pm 0.20$ |
> | UPS                               | $39.47 \pm 0.18$ | $48.29 \pm 0.20$ | $53.27 \pm 0.26$ | $57.83 \pm 0.29$ | $61.16 \pm 0.13$ | $62.58 \pm 0.28$ |
> | VID                               | $37.76 \pm 0.13$ | $46.3 \pm 0.18$  | $52.07 \pm 0.12$ | $57.91 \pm 0.31$ | $60.57 \pm 0.21$ | $63.43 \pm 0.18$ |
> | Weighted                          | $38.39 \pm 0.07$ | $46.91 \pm 0.08$ | $52.48 \pm 0.13$ | $57.7 \pm 0.15$  | $\mathbf{61.32 \pm 0.31}$ | $63.66 \pm 0.46$ |
> | SLaM                              | $\mathbf{41.16 \pm 0.16}$ | $\mathbf{49.11 \pm 0.17}$ | $\mathbf{54.31 \pm 0.19}$ | $\mathbf{58.59 \pm 0.24}$ | $\mathbf{61.30 \pm 0.26}$ | $\mathbf{63.72 \pm 0.43}$ |
>
>
> > In e.g. L230 SLaM is reported to "[...] consistently outperform the baselines, often by a large margin". I believe this is too bold a claim. In particular, only 7 of 24 experiments yield improvements over 1 compared to the second-best baseline, while 8 other of 24 have improvements < 0.2 of which 3 have improvements < 0.02. Furthermore, there are 2 of 24 experiments where the method does not improve over the baselines. All of these observations are without consideration of the variance, which likely makes some additional improvements statistically indistinguishable. I would suggest reducing the claims slightly, especially given the evaluation protocol.
>
> We are certainly open to changing the exact phrasing. However, it is worth mentioning that not all experiments are of the same "importance":
>
> * Experiments in which the the number of available labeled examples is small are of greater importance since this is the typical scenario where "distillation with unlabeled examples" applies.  In these cases, our method indeed often outperforms other methods by a large magin. (See also the "performance gains of SlaM as a Function of the Number of Labeled Examples" section for more details -- bottom of page 8 of our manuscript -- and also the additional experiments we performed to answer the remark of Reviewer Un32.)
> * Our method significantly outperforms the other baselines on Imagenet which is arguably the most "difficult" dataset. As prominent examples: when given labels for only 5% of the Imagenet examples our method is better by **more than 6%** from every other baseline; when given labels for only 10% of Imagenet our method is better by **more than 3%** from every other baseline.
>
>
> Given the above we are willing to change the phrasing to "[...] often by a large margin in the important cases where (i) only few labeled examples are available; (ii) one deals with large-scale problems with many classes like the Imagenet dataset".
>
> We are also happy to adopt any specific suggestion by the reviewers.
>
>
>
> >It is unclear why this method shouldn't be considered a semi-supervised learning method, as both labeled and unlabeled samples are provided and used for the distillation procedure. In fact, the availability of a labeled validations set, V appears to be a necessity for this method, and I believe it should be presented as a semi-supervised setting more explicitly.
>
> * We definitely agree that distillation with unlabeled examples and semi-supervised learning have a lot in common — see the Semi-Supervised Learning paragraph in the "Related Work section" where we explain the similaraties and differences.  It is also true that our method could potentially be applied in a purely semi-supervised learning setting and, moreover, potentially be combined with other semi-supervised learning techniques  — especially given the performance of our methond on Imagenet (a typically difficult dataset for SOTA semi-supervised learning techniques).
> * However, we believe that distillation with unlabeled examples is an important problem of its own (given the popularity of the approach in practical applications) and our goal in this paper is to focus on it and, in particular, to study ways to deal with "teacher's noise" both practically and theoretically.
>
> * We are open to any suggestions by the reviewers for highlighting the connection to semi-supervised learning more explicitly.
>
> > A lot of results and details are deferred to the appendix. While I fully understand the limitations a 9-page restriction can put on a paper, I believe some of the space in the paper could be better utilized to present at least some key results from the appendix.
>
> This is certainly true — and we indeed had a hard time deciding what results we should defer to the appendix. We promise to make an effort to include more results from the appendix to the main body of the paper, and we are open to any suggestions by the reviewers for what they think we should include.

---

> > ### Comment · Reviewer_7BQA · 2023-08-11
> >
> > - Changing to last-epoch evaluations would be an improvement, but I still believe it should be followed by some comments on the improper evaluation scheme. In particular, since I assume the last-epoch evaluation is still based on particular successful experiments, based on the test performance (which should not really be possible beforehand).
> > - The suggested change of phrasing would be adequate for me, however, I would also suggest at least modifying the use of "consistently" as a few experiments are not improved and a few more have negligible improvements (<0.02).
> > - My main concern is that both in the title and the main paper, a lot of emphases is put on the wording "distillation with unlabeled examples", but since the method does require a labeled validation dataset, I believe the naming is misleading.

---

> > > ### Author Response · Authors · 2023-08-13
> > >
> > > Once again we thank the Reviewer for their comments and feedback.
> > >
> > > > Changing to last-epoch evaluations would be an improvement, but I still believe it should be followed by some comments on the improper evaluation scheme. In particular, since I assume the last-epoch evaluation is still based on particular successful experiments, based on the test performance (which should not really be possible beforehand).
> > >
> > > We do understand the Reviewer's concern —  after all, this is a well-known issue that has been raised within the ML community throughout the years. (Unfortunately, the fact is that papers that use standard academic datasets like CIFAR and Imagenet — including every baseline we compared against — do use this evaluation scheme and so we had to follow the same approach to guarantee a fair comparison.)
> > >
> > > To address the Reviewer's concern:
> > >
> > > (i) We will add some comments on the evaluation scheme we used that reflect the Reviewer's concern and our discussion above. (We are happy to adopt any suggestions regarding the phrasing of the latter.)
> > >
> > > (ii) We will include the ablation-experiments which show that our approach is robust to parameters like the size of the validation datasets, the choice of the regression method for learning a(x) and k(x), errors in the estimates of a(x) and k(x) (please see the general response for more details). Hopefully this mitigates the danger of overfitting to the nuances of the academic datasets chosen for evaluation.
> > >
> > > > The suggested change of phrasing would be adequate for me, however, I would also suggest at least modifying the use of "consistently" as a few experiments are not improved and a few more have negligible improvements (<0.02).
> > >
> > > We will modify the use of "consistently" and change our phrasing to:
> > >
> > > "Our approach performs on par with previous SOTA approaches and, interestingly, it often outperforms them by a large margin in the important cases where (i) only a few labeled examples are available; (ii) one deals with large-scale problems with many classes like the Imagenet dataset".
> > >
> > > We are happy to adopt any further suggestions from the reviewers.
> > >
> > > > My main concern is that both in the title and the main paper, a lot of emphases is put on the wording "distillation with unlabeled examples", but since the method does require a labeled validation dataset, I believe the naming is misleading.
> > >
> > > The term "distillation with unlabeled examples" is not ours — it was introduced in [CKSNH] to describe the very popular distillation-setting where one has access to only a few labeled examples but a lot of unlabeled examples.   Note also that using a few of the labeled examples as a validation set is a very common practice in these type of "learning with few labeled examples"-settings and, additionally,  as we mention in Lines 274-278 of the main body of the paper, to be fair to methods not using validation data, we have included the validation data in the training dataset of all methods.
> > >
> > > Another term that has been used to describe the setting we consider is "semi-supervised distillation", but we find it a bit confusing since in standard semi-supervised learning one does not have access to a teacher model.
> > >
> > > Does the reviewer prefer the latter term/have another suggestion?
> > >
> > >
> > >
> > > [CKSNH]: Ting Chen, Simon Kornblith, Kevin Swersky, Mohammad Norouzi, Geoffrey Hinton. Big Self-Supervised Models are Strong Semi-Supervised Learners. NeurIPS 2020.

---

> > > > ### Comment · Reviewer_7BQA · 2023-08-14
> > > >
> > > > I appreciate the reply and effort from the authors toward all reviewers.
> > > >
> > > > The proposed changes do improve the contribution and remove some concerns for me. However, I will keep my current score and suggest for accept.
> > > >
> > > > **"Another term that has been used to describe the setting we consider is "semi-supervised distillation", but we find it a bit confusing since in standard semi-supervised learning one does not have access to a teacher model.
> > > > Does the reviewer prefer the latter term/have another suggestion?"**
> > > > I understand the AI/ML field has an abundance of unclear and duplicate terminology as well as methods with two different namings. However, I believe the inclusion of distillation in "semi-supervised distillation" rightfully states the use of both labeled and unlabeled data as well as a teacher model, whereas "distillation with unlabeled examples" lacks the labeled data reference. Thus, while I personally find "semi-supervised distillation" more accurate, I am not to force this change on you. However, I think this discussion warrants a mention in the paper to avoid any misconceptions about the use of labeled/unlabeled data.

---

> > > > > ### Author Response · Authors · 2023-08-15
> > > > >
> > > > > We would like to thank the reviewer for their suggestion. We will include a discussion on the terms semi-supervised distillation and distillation with unlabeled examples.

---

### Official Review · Reviewer_Un32 · 2023-07-02

**Soundness:** 3 good
**Presentation:** 3 good
**Contribution:** 3 good
**Rating:** 5
**Confidence:** 4

**Summary:**

The paper introduces SLaM, a method for knowledge distillation using unlabeled examples. Its main contribution lies in the development of an efficient and data-agnostic approach that enhances student model performance in scenarios with limited labeled data but abundant unlabeled data. SLaM accomplishes this by incorporating information from both labeled and unlabeled examples during the distillation process. The paper extensively presents experimental results showcasing SLaM's superiority over previous methods on standard benchmarks. Furthermore, the authors provide theoretical guarantees and insights into SLaM's effectiveness. Overall, the paper delivers a valuable contribution to the field of knowledge distillation, showcasing a novel approach supported by empirical evidence.

**Strengths:**

1. The paper introduces a novel approach to knowledge distillation with unlabeled examples that leverages the information from both labeled and unlabeled examples during the distillation process.

2. The experimental results show the effectiveness of the proposed method in knowledge distillation.

3. Theoretical results further guarantee the effectiveness of the proposed method.


**Weaknesses:**

The proposed method argues that existing methods suffer from the shortcoming of noisy pseudo-label, but it also seems hard to guarantee the proposed method can get the pseudo-labels without noisy. Moreover, the proposed method can be limited to semi-supervised learning problem.

**Questions:**

1. Can the proposed method ensure that the pseudo-labels generated by SLaM are noise-free, thereby addressing the limitation discussed in the paper? How and why?

2. Although not specifically designed for knowledge distillation, the baseline method Taylor-CE shows competitive or even superior performance on celebA even with only 2% labeled data. Can you elaborate on the reasoning behind this, especially highlighting the advantages of the proposed method in dealing with sparsely labeled data, which Taylor-CE lacks?

3. It is better to plot a figure to demonstrate the performance of the comparison method in different percents of the labeled training data with a large span. Eg: 5%, 10%, 30%, 50%, 70%.

4. The baseline method Taylor-CE, which focuses on noise label problem, also achieve competitive or even superior performance. I wonder if the proposed method can work in noise label learning that also need reliable pseudo-labels? For example, on clothing1m dataset.

**Limitations:**

Please refer to weakness and question.

---

> ### Author Rebuttal · Authors · 2023-08-10
>
>
> >... but it also seems hard to guarantee the proposed method can get the pseudo-labels without noisy. Can the proposed method ensure that the pseudo-labels generated by SLaM are noise-free, thereby addressing the limitation discussed in the paper? How and why?*
>
>  SLaM effectively improves the quality (reduces the noise) of the pseudo-labels produced by the student. We remark that we "add" noise in the output of the student **only during training** -- during testing/inference no noise is added to the pseudo-labels of the student. While we cannot guarantee that the pseudo-labels generated by a student model trained with SLaM will be perfect (and no other method can guarantee that) we argue that student models trained with SLaM produce better predictions than the available baselines. To show that SLaM effectively reduces the noise of the produced pseudo-labels we perform a self-distillation experiment. We first train a ResNet56 (teacher) on the small labeled Dataset A (its size is shown in the first row of the table below) and then use SLaM to train another ResNet56.  Observe that SLaM produces a student ResNet56 that is more accurate than the teacher ResNet56 for all dataset sizes. Therefore, in this self-distillation experiment the pseudo-labels produced by the student trained with SLaM are **less noisy** than the labels provided initially by the teacher.
>
>
> | CIFAR100 (Resnet 56->56) |   1%    |   5%   |   10%   |   15%   |  20%   |  25%   |  30%   |  35%   |  50%   |  60%    |
> |-|-|-|-|-|-|-|-|-|-|-|
> | Teacher                  |   8.78   |   21.06  |   34.15  |   42.72  |   47.9   |   53.47  |   56.41  |   58.73  |   64.4   |   66.35  |   69.18  |
> | Vanilla                  | $9.36 \pm 0.03$  | $22.42 \pm 0.11$ | $36.34 \pm 0.30$ | $44.93 \pm 0.34$ | $50.74 \pm 0.36$ | $55.8 \pm 0.11$  | $58.72 \pm 0.33$ | $61.24 \pm 0.15$ | $66.43 \pm 0.04$ | $68.4 \pm 0.02$  |
> | Taylor-CE                   | $9.36 \pm 0.05$  | $23.58 \pm 0.15$ | $38.41 \pm 0.19$ | $46.45 \pm 0.09$ | $52.53 \pm 0.11$ | $56.67 \pm 0.31$ | $59.79 \pm 0.23$ | $61.98 \pm 0.31$ | $66.53 \pm 0.35$ | 68.43+/-0.30 |
> | SLaM            | $\mathbf{10.52 \pm 0.4}$ | $\mathbf{25.66 \pm 0.2}$ | $\mathbf{39.72 \pm 0.3}$ | $\mathbf{48.46 \pm 0.4}$ | $\mathbf{53.53 \pm 0.15}$ | $\mathbf{57.38 \pm 0.31}$ | $\mathbf{60.64 \pm 0.13}$ | $\mathbf{62.27 \pm 0.32}$ | $\mathbf{66.60 \pm 0.15}$ | $\mathbf{68.47 \pm 0.24}$  |
>
>
> >... Taylor-CE shows competitive or even superior performance on celebA even with only 2% labeled data. Can you elaborate on the reasoning behind this, especially highlighting the advantages of the proposed method ... ?*
>
> * Taylor-CE outperforms SLaM only for the 2\% case of the Celeb-A dataset.  In contrast, SLaM outperforms Taylor-CE in all other experiments and in some cases by a large margin (see, e.g., the 5\% case of **Imagenet at Table 4 where SLaM outperforms Taylor-CE by ~6.5\%** and the CIFAR100 ablation of the previous answer).
>
>
> * SLaM almost always outperforms Taylor-CE because the noise in the teacher's labels is structured (not random nor adversarial) and SLaM exploits this structure while Taylor-CE does not (was designed for generic noisy labels).
>
> * SLaM can be combined with different loss functions (we used the cross-entropy in our experiments because it is the most commonly used one).  In fact, in Section D.6 and Figure 7 of the supplementary material we show that SLaM can be used with the Taylor-CE loss function and improve its performance.
>
>
> > It is better to plot a figure to demonstrate the performance of the comparison method in different percents of the labeled training data with a large span. Eg: 5%, 10%, 30%, 50%, 70%.
>
> * We included smaller labelled training data (1%, 5%) for CIFAR10/100.  We observe that SLaM consistently outperfoms the baselines. For CIFAR10 with 1% labelled datas SLaM achieved an improvement of over 10% over other baselines. For the results, we refer to Item 1 of  Additional Experiments in our general response.
>
> * For larger dataset sizes, as seen in the plots of  our manuscript, all methods converge to roughly the same performance (the setting is more similar to standard knowledge distillation where the teacher model has access to the full labelled dataset). See also our ablation in the previous question.
>
> * Experiments in which the the number of available labeled examples is small are of greater importance since this is the typical scenario where "distillation with unlabeled examples" applies.  (See also the "performance gains of SlaM as a Function of the Number of Labeled Examples" section -- bottom of page 8 of our manuscript -- for more details, and also the additional experiments we performed to answer the remark of Reviewer 7BQA.)
>
>
> >I wonder if the proposed method can work in noise label learning that also need reliable pseudo-labels? For example, on clothing1m dataset.
>
> SLaM is related to forward methods for dealing with label noise (see our Related Work section for more details).  However, the fundamental difference between our setting and the general learning under label noise setting is that the noise introduced by the teacher is structured, and this is a crucial observation we utilize in our design. Specifically, our approach is specifically tailored to the structure of the distillation with unabeled examples setting by exploiting that (i) we have access to confidence metrics of the teacher's predictions; and (ii) that often times, when the teacher model's top-$1$ prediction is inaccurate the true label is within its top-$k$ predictions for some appropriate $k$, to design and estimate a much more refined model for the teacher's noise that we use to inform the design of the student's loss function.  Given the above SLaM cannot be applied to general learning from noisy-labels tasks but it is interesting to investigate whether adaptations of it could be helpful in those settings.

---

> > ### Comment · Reviewer_Un32 · 2023-08-16
> >
> > Thanks for the authors’ response.
> >
> > **To A1**, the author said the pseudo-labels produced by the proposed method can also be noisy, but the author motivates the new method due to existing methods suffering from the shortcoming of noisy pseudo-label. It seems the proposed method does not address the shortcoming of the existing method.
> >
> > Moreover, from the comparison result below, we can see that the benefits can be marginal, which demonstrates the proposed method is not substantially different from the comparison method.
> >
> > **To A2**, the author does not answer my question Q2. SLaM beats Taylor-CE in other experiments cannot explain why the baseline method Taylor-CE beats the proposed method.

---

> > > ### Author Response · Authors · 2023-08-16
> > > **Clarifications needed**
> > >
> > > >To A1, the author said the pseudo-labels produced by the proposed method can also be noisy, but the author motivates the new method due to existing methods suffering from the shortcoming of noisy pseudo-label. It seems the proposed method does not address the shortcoming of the existing method.
> > >
> > > We show that student models trained with SLaM will produce **better** pseudo-labels than other methods.  **It is impossible to generate perfect pseudo-labels and no method achieves that**.
> > >
> > > Perhaps the Reviewer could clarify what they mean by “suffering from the shortcoming of noisy pseudo-label” and “It seems the proposed method does not address the shortcoming of the existing method.”
> > >
> > > >To A2, the author does not answer my question Q2. SLaM beats Taylor-CE in other experiments cannot explain why the baseline method Taylor-CE beats the proposed method.
> > >
> > >
> > > In our previous response we tried to explain that, since Taylor-CE only beats our method in 1 out of the 28 experiments we considered, we feel that there is no trend to be explained here —  that particular data-point is most probably a random outlier.
> > > In general, the advantage of our method compared to Taylor CE (and any other generic label-noise method) is that our method is designed to provably exploit the structure of the instance specific noise from the teacher model — this is why it outperforms Taylor CE  in 27 out of the 28 experiments, and by a large margin in the most difficult dataset, i.e.,  Imagenet.
> > >
> > > We feel that we may have not understood the Reviewer’s question (because, frankly, we see no trend justifying that Taylor CE in particular is better than our method), so maybe the Reviewer could further clarify what they are asking?
> > >
> > >
> > > >Moreover, from the comparison result below, we can see that the benefits can be marginal, which demonstrates the proposed method is not substantially different from the comparison method.
> > >
> > >
> > > We respectfully disagree with the reviewer: the improvements are **not marginal** as  there are several cases where SLaM outperforms baselines by large margins (sparsely labeled data cases), while all methods converge to the same performance as the noise from the teacher is decreasing (as expected). See the paragraph “Performance Gains of SLaM as a Function of The Number of Labeled Examples “ for more details.

---

### Official Review · Reviewer_PQzZ · 2023-07-02

**Soundness:** 4 excellent
**Presentation:** 4 excellent
**Contribution:** 3 good
**Rating:** 7
**Confidence:** 3

**Summary:**

In their paper, the authors propose a knowledge distillation procedure that augments pseudo-labels predicted by a teacher models by its uncertainty, leading to a uncertainty-aware student learning routine. This is realized by a novel loss formulation that mixes student predictions with quantified uncertainty associated with the pseudo-labeling prediction. Together with convincing empirical results demonstrating improvements in terms of generalization performance, theoretical results characterize the convergence of the proposed solution.

**Strengths:**

- Strong empirical results against recent baselines
- Profound theoretical results
- Well-written manuscript with a clear and valid motivation, addressing a relevant problem
- The simplicity of the approach is appealing, promoting the applicability of this method


**Weaknesses:**

Major:

- While the empirical evaluation itself is certainly convincing from a generalization point of view, a more rigorous ablation study of the influence of the uncertainty quantification would be insightful. Namely, it is hard to judge how sensible this procedure is towards a good quality of the isotonic regression solution. One possibility to address this concern could be by obtaining $\alpha(x)$ and $k(x)$ from an oracle with “perfect” estimations, and gradually contaminating these estimates. Then, one could see how the performance evolves.
- Relating to the previous comment, I am wondering why exactly isotonic regression is chosen as a means to quantify $\alpha(x)$. I see little explanations on why exactly this method is chosen, and not others, like conformal prediction. Of course, it is completely valid to make an arbitrary design choice that works well, but I would love to see a more thorough elaboration on the motivation of this specific construction. An ablation study as described before could be used to support this decision.

Minor:

- The figures are relatively small and hard to read, enlarging them would be appreciated
- The orthography could be improved, many missing commas (e.g., when starting sentences like “In Section xyz, …”)
- The references are inconsistent (e.g., the conference identifiers: NeurIPS, ICML, … vs. International Conference on Learning Representations)


**Questions:**

-

**Limitations:**

I do not see too much about limitations, this could certainly be deepened.

---

> ### Author Rebuttal · Authors · 2023-08-10
>
>
>
> >.. a more rigorous ablation study of the influence of the uncertainty quantification would be insightful. ... obtaining $\alpha(x)$ and $k(x)$ from an oracle with “perfect” estimations, and gradually contaminating these estimates.
>
> We agree with the reviewer that a "controlled" experiment by adding noise to test the robustness of SLaM to inaccurate predictions of $\alpha(x)$ and $k(x)$ is a valuable addition to our experimental evaluation.  To do this, as the reviewer suggested, we start from the oracle values for $\alpha(x)$ and $k(x)$, i.e., $\alpha^*(x) = 1$ if the teacher prediction is correct on $x$ and $0$ if it is incorrect and $k^*(x)$ is equal to the smallest integer value $\ell$ so that the ground-truth label is contained in the top $\ell$ predictions of the teacher. We then introduce random noise to the oracle predictions: for $\alpha(x)$ we perform a random perturbation $$\alpha(x) = \alpha^*(x) + (1-2\alpha^*(x)) ~ \xi, $$ where $\xi$ is uniformly distributed in $[0,\sigma]$. Hence, when $\alpha^*(x) = 0$ we increase it by adding a random variable $\xi \in [0,\sigma]$ and when $\alpha^*(x) = 1$ we decrease it by subtracting the same random variable $\xi \in [0,\sigma]$. We clip the resulting value in the interval $[0,1]$.
> To create noisy values for $k(x)$ we simply add a random integer to the optimal value $k^*(x)$, i.e., $k(x) = k^*(x) + Z$, where $Z$ is a random integer in $\{-\ell,\ldots,\ell\}$. We clip the resulting value of $k(x)$ in the range $\{0,\ldots, numClasses\}$.
>
> The table with our results on CIFAR100 can be found in our general response (Item 2 of Additional Experiments).
>
> >... why exactly isotonic regression is chosen ...  An ablation study as described before could be used to support this decision.
>
> We chose isotonic regression to enforce the monotonicity in the learned accuracy estimates for $\alpha(x)$ based on the empirical observation that $\alpha(x)$ is often approximately monotone as a function of the margin of the teacher. Moreover, the lower threshold (denoted by lb in Section B.1. in the Supplementary Material) in isotonic regression gives us a way to control how "agressive" the mixing operation is going to be.    We agree with the reviewer that SLaM does not hinge on some particular regression method and other methods can be used.
> We used SLaM with the k-nearest neighbors regression and with logistic regression and found that isotonic regression typically outperfoms logistic regression and its hyper-parameter is easier to tune than the number of neighbors in kNN.  That said, SLaM is agnostic to the methods for estimating $\alpha(x), k(x)$ and more sophisticated methods (such as conformal prediction) could also be used.  We will include this discussion and ablations in our updated manuscript.
>
> | CIFAR10              | 10%           | 15%           | 20%           | 25%           | 30%           | 35%           |
> |-|-|-|-|-|-|-|
> |kNN  k=10  | $67.1 \pm 0.15$ | $70.56 \pm 0.21$ | $74.6 \pm 0.11$ | $76.68 \pm 0.17$ | $78 \pm 0.16$    | $79.26 \pm 0.12$ |
> | kNN  k=20  | $67.5 \pm 0.15$ | $71.09 \pm 0.19$ | $74.47 \pm 0.15$ | $77.03 \pm 0.1$  | $78.03 \pm 0.17$ | $79.21 \pm 0.13$ |
> |kNN  k=30  | $67.51 \pm 0.11$| $71.27 \pm 0.13$| $74.66 \pm 0.12$| $77.03 \pm 0.13$| $78.07 \pm 0.2$ | $79.2 \pm 0.18$  |
> |kNN  k=40  | $67.64 \pm 0.21$| $71.08 \pm 0.22$| $74.5 \pm 0.09$ | $76.64 \pm 0.12$| $77.92 \pm 0.11$| $79.41 \pm 0.22$ |
> |Logistic | $65.26 \pm 0.05$ | $68.85 \pm 0.08$ | $73.35 \pm 0.12$ | $76.17 \pm 0.15$ | $76.87 \pm 0.25$ | $78.76 \pm 0.35$|
> |Isotonic lb=0.5| $66.82 \pm 0.61$ | $72.61 \pm 0.30$ | $75.01 \pm 0.25$ | $75.72 \pm 0.17$ | $78.04 \pm 0.16$ | $79.22 \pm 0.11$|
>
> | CIFAR100 |  10%    |  15%    |  20%    |  25%    |  30%    |  35%    |
> |-|-|-|-|-|-|-|
> | kNN k=10  | $40.75 \pm 0.02$ | $49.07 \pm 0.15$ | $54.86 \pm 0.11$ | $57.87 \pm 0.17$ | $61.9 \pm 0.2$  | $63.06 \pm 0.22$ |
> | kNN k=20  | $41.03 \pm 0.05$ | $49.19 \pm 0.12$ | $54.9 \pm 0.1$   | $57.85 \pm 0.18$ | $61.76 \pm 0.2$  | $63.45 \pm 0.16$ |
> | kNN k=30  | $41.04 \pm 0.07$ | $49.55 \pm 0.13$ | $55.14 \pm 0.15$ | $57.96 \pm 0.21$ | $61.9 \pm 0.19$  | $63.2 \pm 0.23$  |
> | kNN k=40  | $41.23 \pm 0.03$ | $49.76 \pm 0.15$ | $54.78 \pm 0.1$  | $58.15 \pm 0.17$ | $61.98 \pm 0.21$ | $63.47 \pm 0.2$  |
> | Logistic | $39.68 \pm 0.03$ | $48.17 \pm 0.1$  | $53.56 \pm 0.11$ | $57.45 \pm 0.09$ | $61.77 \pm 0.19$ | $63.24 \pm 0.18$ |
> |Isotonic lb=0.5 | $42.72 \pm 0.30$  |  $49.89 \pm 0.23$ | $54.73 \pm 0.27$ | $58.78 \pm 0.15$ | $61.30 \pm 0.09$ | $63.98 \pm 0.19$|
>
> >.. Minor comments regarding presentation
>
> We will implement all the reviewer's suggestions regarding the presentation of our paper.

---

> > ### Comment · Reviewer_PQzZ · 2023-08-13
> >
> > Thanks for the author's response, and the efforts spent into further complementing the insights. The additions improve the overall contribution, which makes me even more confident in proposing to accept this paper.

---

> > > ### Author Response · Authors · 2023-08-13
> > >
> > > We greatly appreciate the Reviewer's feedback and effort in evaluating our work. We will adopt all of their suggestions and add remarks that summarize our discussion.

---

### Official Review · Reviewer_E5Wk · 2023-07-08

**Soundness:** 3 good
**Presentation:** 3 good
**Contribution:** 3 good
**Rating:** 6
**Confidence:** 3

**Summary:**

The paper attempts to resolve the problem of noisy teacher labels for unlabeled examples in knowledge distillation. Proposed Student-Label Mixing (SLaM) to introduce noise into student prediction when matching to teacher’s  pseudo-label on unlabeled data. Experiments on standard benchmarks were used to demonstrate the effectiveness of the proposed method. Also, the paper provided theoretical guarantees for SLaM for learning noisy halfspaces.

rating updated after author rebuttal.

**Strengths:**

The paper studied the critical problem about how to learn unlabeled examples for the student model with knowledge distillation when the teacher model is not perfect.

The paper proposed to resolve it from a novel angle by introducing noise to the student model’s prediction, and also provided a theoretical guarantee for the proposed solution.

Extensive experiments based on common datasets and multiple baselines were provided to demonstrate its effectiveness.


**Weaknesses:**

The paper lacks the explanation on the intuition of the mixing solution on formula (1), where there is no clarification why it is essential for the second term, i.e. the element-wise multiplication term.

Even though the paper developed a solution to consider the noise for student model prediction, it seems not robust enough to learn \alpha(x) and k(x). As it requires an additional validation dataset, it is easy to make \alpha overfit to those data. Also it has another strong assumption that \alpha is ​​isotonic regression. All of these make the learning of \alpha easily overfit the data which cannot be robust enough, which also make the solution not that elegant.

It lacks the result details about \alpha and k setup for each experiment. Also it lacks the related ablation study to demonstrate how \alpha and k affect the model performance.

no comparison with a baseline where the validation dataset is used for vanilla distillation, and on top that add vanilla unlabeled distillation or other state of the art unlabeled distillation.


**Questions:**

The proposed algorithm has the additional cost of needing a validation dataset, seems a fair baseline would be to incorporate the validation set for vanilla distillation, and on top that add vanilla unlabeled distillation or other state of the art unlabeled distillation?

What is the intuition of the second term on formula (1)?

How important is the design \alpha(x) and k(x) is? How does the size of the validation set affect the overall performance?


**Limitations:**

yes

---

> ### Author Rebuttal · Authors · 2023-08-10
>
> > The paper lacks the explanation on the intuition of the mixing solution on formula (1), w ... *What is the intuition of the second term on formula (1)?
>
> **Short Answer**
> * The high-level intuition of equation (1) appears in Lines 212-226 of the manuscript.
> * In Appendix C of the supplementary material we have provided a **formal proof** that the minimizer of the SLaM objective using equation (1) is the ground-truth labeling (given noisy teacher predictions).
> * The second term is needed to ensure that the minimizer of the SLaM objective is the ground-truth labeling, see Appendix C. The expected noisy teacher label is equal to $\alpha(x) g(x) + (1-\alpha(x)) (1-g(x))$ (where g(x) is the ground-truth label of x).The noisy student is equal to $\alpha(x) f(x) + (1-\alpha(x)) (1-f(x))$. When the noisy teacher matches the noisy (mixed) student it must be that $f(x) = g(x)$ (the student matches the ground-truth). See  Appendix C for more details.
>
> > Even though ... not robust enough to learn $\alpha(x)$ and $k(x)$. A... it is easy to make $\alpha$ overfit to those data.  -- How important is the design $\alpha(x)$ and $k(x)$ is?
>
> * As we mention in our manuscript, the parameter $\alpha(x)$ that we estimate is only a function of the teacher's margin so it can be learned from very few data via a **simple one dimensional** regression task.
> * SLaM is robust to noise in the estimates of $\alpha(x)$ and $k(x)$; only rough estimates are needed for SLaM to provide improvements. In fact, in many cases, **even using a single value for $k(x)$ for all examples (say k(x)=5) is enough for SLaM to achieve good performance** (see our Imagenet experiments in Table 4 and also our response below). We also test the robustness of SLaM to noise in the predictions of $\alpha(x)$ and $k(x)$ in the ablation provided to Reviewer PQzZ.
>
> >Also it has another strong assumption that \alpha is isotonic regression.
>
> **Using isotonic regression is not an assumption about $\alpha(x)$**: our regression algorithm works regardless of whether $\alpha(x)$ is monotone as a function of the margin.
>  As we mention in Figure 1 and Lines 564-571, using isotonic regression is just a way to enforce the monotonicity of the resulting estimator — a property we have empirically observed that enhances the robustness of SLaM and leads to more stable estimates of $\alpha(x)$ (*harder to overfit*) -- see also our general response and the response to Reviewer PQzZ.
>
> >It lacks the result details about \alpha and k setup for each experiment. Also it lacks the related ablation study ....
>
> **Short Answer** Due to space limitations, the details for $\alpha(x)$ and $k(x)$ were included in section D of the supplementary material.  See Lines 677-680 for Celeb-A, Cifar10/100, Lines 689-693 for Imagenet, Lines 726-727 for Large Movies Reviews Dataset.
>
> **More Details**
>
> Binary Classification. Our method only requires tuning a single hyperparameter: the accuracy lower bound for a(x) in isotonic regression, see the value lb in the first paragraph in Section B.1.
>
> The effect of the parameter lb is clear from the mixing operation (see the mixing operation above Equation 2 in Section 3). As the accuracy lower bound (lb) converges to 1, SLaM converges to “vanilla” distillation as we do not mix the student’s predictions. When the accuracy lower bound is closer to 0, SLaM more aggressively mixes student’s predictions.
>
> Multiclass Classification. For the isotonic regression lower bound in CIFAR-10 and CIFAR100, we used lb=0.5. We found out that SLaM is robust in the setting of lb in those datasets and the values {0.5,0.6,0.7} usually yield good performance. See ablation in Table 2 of the pdf.
>
> We use two methods to obtain k(x).
>
> The first one uses the same value k(x) = k for all examples. We found that it suffices to achieve good performance gains. We used this method in our ImageNet experiments and used the value k=5 as the top-5 accuracy of the teacher model was satisfactory (much higher than its top-1 accuracy) on the validation dataset.
>
> The second way uses a (data-dependent) value of k(x) for each example (see Sections B.1. and B.2.). This requires tuning a hyperparameter t (see the top-k accuracy threshold in Algorithm 2). We used this method in CIFAR10/100 and used t=0.9.
>
> In Table 3 of the rebuttal pdf, we present our results for CIFAR100 using a fixed value for k(x). We observe that SLaM is rather robust to the value of k used since it outperforms vanilla distillation for some reasonable values of k. Overall, we found that using fixed values of $k(x)$ (after some hyper-parameter search for $k$) and using the data-dependent method yield comparable results. The advantage of the fixed-value method is that it is easier to implement (and slightly more efficient) and the advantage of the data-dependent method is that its hyperparameter (threshold $t$ in Algorithm 2) is easier to tune (in all our experiments $t=0.9$ achieved good performance).
>
> >no comparison with a baseline where the validation dataset is used ... incorporate the validation set for vanilla distillation ...?
>
> Αs we mention in Lines 274-278 of the main body of the paper, to be fair to methods not using validation data, we have included the validation data in the training dataset of all methods compared.
>
> >How does the size of the validation set affect the overall performance?
>
> As we already discussed SLaM requires only rough estimates of $\alpha(x)$ and $k(x)$ and thus even very small validation datasets suffice.  In Table 1 of the rebuttal pdf we use different validation sizes for the CIFAR-100 experiment described in our manuscript and and show that the performance of SLaM improves when the validation dataset is larger but the gaps are not very significant especially for larger sizes of the labeled dataset. We show that SLaM is able to provide improvements even with very small validation datasets (e.g., with 128 labels).  See also Item 2 of Additional Experiments in the general response.

---

> > ### Comment · Reviewer_E5Wk · 2023-08-17
> > **thanks for addressing the questions**
> >
> > thanks for addressing the questions. I increased my score.

---

> > > ### Author Response · Authors · 2023-08-17
> > > **Ack**
> > >
> > > Thank you for carefully going over our response!

---

### Author Rebuttal · Authors · 2023-08-10


## General Response
We want to thank all reviewers for taking the time to read our manuscript carefully and for providing constructive and insightful feedback. We are very encouraged by the positive comments of the reviewers on **tackling a well-motivated and important problem** (Reviewers E5Wk, PQzZ) the **simplicity, generality, and novelty of SLaM** (Reviewers E5Wk, PQzZ, Un32, 7BQA), **its extensive experimental evaluation and performance gains** (Reviewers E5Wk, PQzZ, Un32), **its strong theoretical guarantees** (Reviewers PQzZ, Un32, 7BQA), **the writing quality and the clarity of the presentation of the ideas** (Reviewers PWzZ, 7BQA).

Here we give an update about our theoretical contribution and provide an overview of the additional experiments included in the rebuttal. We provide detailed responses to each reviewer separately.

### Additional Experiments

1. **SLaM with (even) fewer labels**
As proposed by reviewer Un32, we extended the range of the the size of the labeled-dataset considered in our experiments. In particular, we provide experiments with even fewer labeled examples available. (Experiments in which the number of available labeled examples is small are of greater importance since this is the typical scenario where "distillation with unlabeled examples" applies. For larger dataset sizes, as seen in the plots of  our manuscript, all methods converge to roughly the same performance.) See response to Reviewer Un32 for more details.

|Method||CIFAR10 1%|CIFAR10 5%||CIFAR100 1%|CIFAR100 5%|
|-|-|-|-|-|-|-|
|Teacher | |$10.07$| $51.67$ || $9.45$ | $23.61$||
|Vanilla || $11.75 \pm 0.1$| $54.2 \pm 0.16$|| $10.08 \pm 0.06$| $25.15 \pm 0.11$|
|Taylor-CE || $10.00 \pm 0.01$| $55.14 \pm 0.28$|| $9.79 \pm 0.13$| $26.14 \pm 0.39$|
|UPS|| $12.74 \pm 0.94$| $56.21 \pm 0.23$| | $10.40 \pm 0.05$| $26.41 \pm 0.13$|
|VID|| $13.25 \pm 0.26$| $54.32 \pm 0.05$|| $10.02 \pm 0.13$| $24.93 \pm 0.19$|
|Weighted || $12.67 \pm 0.01$| $54.58 \pm 0.1$ || $10.07 \pm 0.06$| $25.36 \pm 0.14$|
|SLaM || $\mathbf{26.73 \pm 0.02}$| $\mathbf{57.40 \pm 0.05}$|| $\mathbf{10.87 \pm 0.07}$| $\mathbf{27.76 \pm 0.19}$|

2. **Robustness of SLaM to inaccuracies in $\alpha(x), k(x)$**
We tested the robustness of SLaM to inaccurate predictions for $\alpha(x), k(x)$ as proposed by Reviewer PQzZ.  As the reviewer suggested, we start from the oracle values for $\alpha(x)$ and $k(x)$, i.e., $\alpha^*(x) = 1$ if the teacher prediction is correct on $x$ and $0$ if it is incorrect and $k^*(x)$ is equal to the smallest integer value $\ell$ so that the ground-truth label is contained in the top $\ell$ predictions of the teacher. We then introduce random noise to the oracle predictions: for $\alpha(x)$ we add random perturbations with variance roughly $\sigma^2$ and for $k(x)$ we add a random integer in the interval $[-\ell, \ell]$ to the oracle prediction $k^*(x)$.  For more details on the noise operation we refer to our response to Reviewer PQzZ. We test our method on CIFAR100 and **observe that the accuracy of SLaM gracefully decays as the predictions for $\alpha(x)$ and $k(x)$ become worse** (bottom right corner is the noisiest $\sigma = 0.5, \ell=90$ and top left is the noiseless $\sigma=0, \ell =0$).


|CIFAR100 20% Labeled Data | $\ell$ = 0|$\ell$ = 5| $\ell$ = 10 | $\ell$ = 50 | $\ell$ = 90|
|-|-|-|-|-|-|
|$\sigma$ = 0| $61.69 \pm 0.3$| $59.71 \pm 0.49$ | $59.7 \pm 0.55$  | $59.63 \pm 0.44$ | $59.4 \pm 0.65$ |
|$\sigma$ = 0.1| $60.04 \pm 0.29$| $59.55 \pm 0.33$ | $59.66 \pm 0.35$ | $59.79 \pm 0.2$  | $60.03 \pm 0.41$|
|$\sigma$ = 0.2| $59.39 \pm 0.1$ | $59.23 \pm 0.25$ | $59.5 \pm 0.39$  | $59.16 \pm 0.29$ | $59.31 \pm 0.31$|
|$\sigma$ = 0.5| $57.71 \pm 0.15$ | $57.6 \pm 0.23$  | $57.56 \pm 0.28$ | $57.46 \pm 0.25$ | $57.29 \pm 0.21$|

3. Ablations for hyperparameters of $\alpha(x)$ and $k(x)$ as requested by reviewer E5Wk.  We investigated the effect of the isotonic-regression lower bound hyper-parameter and the two methods for estimating $k(x)$. For more we refer to our response to Reviewer E5Wk. The corresponding tables are Table 2 and Table 3 in the pdf of the rebuttal.

4. We investigated using different regression methods (knn, Logistic regression) for estimating $\alpha(x), k(x)$ with SLaM as proposed by Reviewer PQzZ.  **We see that Isotonic regression typically outperfoms other methods. Moreover, SLaM provides consistent improvements regardless of regression method used**.  We refer to our response to Reviewer PQzZ for more details.

5. An ablation investigating the effect of the validation size, see Response to Reviewer E5Wk. **We show that SLaM is able to provide improvements even with very small validation datasets** (e.g., with 128 labels). This is consistent with the fact that SLaM is robust to inaccuracies in the parameters $\alpha(x), k(x)$, see Item 2 of Additional Experiments in the general response.

6. We include a table reporting the last-epoch accuracy of all methods on CIFAR100 as proposed by Reviewer 7BQA. The last-epoch accuracy of SLaM is again better than the baselines. See our response to Reviewer 7BQA for more details.

### Theoretical Contribution

We would like to share with the reviewers and ACs an exciting recent result presenting strong evidence about the optimality of our theoretical result for SLaM.

In the very recent paper [DKK+23] a Statistical Query (SQ) lower bound of $\Omega(1/(\gamma^{1/2} \epsilon^2))$ was given for learning halfspaces with random classification noise. This implies that there is now strong evidence (SQ algorithms is very broad class that contains gradient-based optimization methods) that **our theoretical result for SLaM is, in fact, optimal in its dependence on the accuracy parameter $\epsilon$.**  We will update our conclusion section to include this discussion.

[DKK+23]: Diakonikolas I., Diakonikolas J., Daniel K., Wang P., Zarifis N. Information-Computation Tradeoffs for Learning Margin Halfspaces with Random Classification Noise. COLT 2023

---

### Decision · Program_Chairs · 2023-09-21

**Decision:**

Accept (poster)

**Comment:**

This paper proposes a method for improving knowledge distillation with unlabeled examples called Student-Label Mixing (SLAM). Knowledge distillation is widely used, and this paper seeks to improve the process by modeling the noise from the teacher model. By evaluating the student model using both its prediction and an estimated label from the noise model, SLAM can reduce the effect of noise in the training pseudo-labels. Experiments show that SLAM provides consistent performance improvements on standard benchmarks and beneficial combination with other standard techniques for distillation. A theoretical analysis also shows that SLAM can provably learn halfspaces with noise with sample complexity based on the size of the margin and amount of noise.

During the review and discussion period, the reviewers identified the contributions to knowledge distillation as novel and valuable. The principled approach was viewed as a strength. The main concerns of the reviewers generally focused on the experimental evaluation. Two areas that came up were evaluating SLAM on even smaller amounts of labeled data and evaluating its robustness by introducing additional noise. The authors provided additional results addressing these concerns and the reviewers were all in favor of acceptance. The authors are strongly encouraged to add these additional results to the final version of the paper.